# An RNA sponge directs the transition from feast to famine in *Caulobacter crescentus*

Laura N. Vogt [1,3], Manuel Velasco Gomariz [1,3], Malte Siemers [1,2], Kai Papenfort [1,2] & Kathrin S. Fröhlich [1,2] ✉

In bacteria, regulatory networks controlling the adaptation of gene expression in response to stress are frequently complemented by base-pairing small regulatory RNAs (sRNAs) that act at the post-transcriptional level. While many regulatory circuits governing stress resilience have been studied in the model bacterium *Caulobacter crescentus*, only a small fraction of its diverse sRNA repertoire has been characterized. In this study, we globally identify interacting RNA-RNA pairs associated with the major RNA-binding protein Hfq in *C. crescentus*. In addition to numerous connections between sRNAs and mRNAs, we also recover RNA-RNA pairs consisting of two non-coding transcripts. Our results indicate that the sRNA CrfA acts as a sponge to inactivate a family of four conserved sRNAs, SisA-D. When induced by carbon starvation, CrfA redirects gene expression towards the utilization of distinct energy sources, and loss of the RNA sponge is linked to a severe growth defect in environments with fluctuating nutrient availability.

Bacteria experience tremendous heterogeneity in their micro-environments. The spatial and temporal variation of key parameters within a habitat, such as nutrient availability, ion concentrations, or temperature, enforces constant re-organization of their gene expression profiles to optimize fitness and to manifest survival[1-3]. The Alphaproteobacterium *Caulobacter crescentus* is a prominent and genetically tractable model to study microbial assimilation to resource-limited habitats[4]. As the freshwater environment naturally populated by *C. crescentus* (hereafter: *Caulobacter*) typically contains a dilute, yet complex composition of nutrients, the bacterium is highly adapted to an oligotrophic diet but is also able to cope with phases of starvation[5,6]. Abrupt carbon depletion results in the global rewiring of *Caulobacter* gene expression programmes, including blockage of DNA replication and cell growth[7,8]. Simultaneously, the lack of primary carbohydrates also activates numerous genes, many of which are predicted to facilitate the decomposition of plant-derived carbon sources abundant in the freshwater environment[9,10]. This focus on different nutrient sources is further reflected in a reorganization of the outer membrane protein profile, including the upregulation of more than a dozen TonB-dependent receptors (TBDRs). TBDRs facilitate the selective transport of substrates across the outer membrane by harnessing the proton motive force generated by the TonB-ExbB-ExbD complex (the TonB system) in the inner membrane[11,12], and are likely utilized for the import of plant polysaccharide breakdown products in the response to starvation. A previous study revealed the requirement of the CrfA (*Caulobacter* response to famine) small RNA (sRNA) for the carbon starvation-dependent induction of at least 27 genes, encoding proteins involved in intermediary nutrient metabolism, as well as TBDRs[13].

Bacterial sRNAs constitute an abundant and structurally heterogeneous class of post-transcriptional regulators of gene expression[14-16]. In *Caulobacter*, more than 150 sRNA candidates have been annotated; however, the majority still await functional characterization[17-19]. When induced, sRNAs act on their target mRNAs through limited base-pairing interactions, which can be promoted by the RNA-binding protein Hfq[20,21]. Hfq forms homohexamers with multiple RNA binding sites to simultaneously interact with sRNAs and mRNAs, and – given sufficient sequence complementarity – facilitates their annealing[22,23]. Recognition by an sRNA typically occurs within the mRNA's 5′ untranslated region (5′ UTR) and can affect transcript

[1]Institute of Microbiology, Faculty of Biological Sciences, Friedrich Schiller University, Jena, Germany. [2]Cluster of Excellence Balance of the Microverse, Friedrich Schiller University, Jena, Germany. [3]These authors contributed equally: Laura N. Vogt, Manuel Velasco Gomariz.
✉e-mail: kathrin.froehlich@uni-jena.de

stability, translation, or both[20,21]. Commonly, sRNAs inhibit target mRNA expression through occupation of the translation initiation site or recruitment of the RNA decay machinery[24]. Alternatively, an sRNA may function as an activator of target gene expression if annealing blocks the activity of negative regulators or interferes with a self-inhibitory secondary structure of an mRNA[25]. The implementation of methodologies that comprehensively capture RNA-RNA interactomes has greatly aided the identification of sRNA target spectra and concomitantly, the assignment of individual sRNAs to regulatory networks[15,26,27]. In addition, these protocols promote the recovery of RNA-RNA interactions, including sponge RNAs that counteract regulatory sRNAs through sequestration[28–30]. Initially identified in eukaryotes[31,32], the continually increasing number of RNA sponges discovered in bacteria is indicative of their crucial role for base-pairing competition in post-transcriptional regulation of prokaryotes. Sponge RNAs can be produced in different ways. In some cases, these regulators originate from decay intermediates of functional transcripts, as is the case for the tRNA fragment 3'ETS$^{leuZ}$ (antagonizing RyhB and RybB sRNAs)[33] or SroC, which is processed from *gltIJ* mRNA (antagonizing GcvB sRNA)[34]. RNA sponges may also be transcribed from independent genes, e.g., the phage-encoded AsxR and AgvB (antagonizing FnrS and GcvB sRNAs, respectively)[35].

In this study, we have revealed the sponging activity of the carbon starvation-induced sRNA CrfA in *Caulobacter*. We used RIL-seq (RNA interaction by ligation and sequencing)[36], which relies on the co-immunoprecipitation of RNA duplexes together with the RNA chaperone Hfq, to identify global RNA networks. Our approach uncovered hundreds of previously unknown RNA-RNA pairs, including those involving CrfA. Contrary to previous findings[13], we discovered that CrfA does not interact with its proposed target mRNAs, but instead functions as an RNA sponge to inactivate four homologous sRNAs of the αR8 RNA family, SisA-D (sponge-interacting sibling sRNA A-D). We show that under nutrient-rich conditions, the highly abundant SisA sRNA post-transcriptionally represses a large regulon previously suggested to be directly controlled by CrfA. When carbon becomes scarce, CrfA is induced and sequesters SisA through an Hfq-mediated base-pairing interaction, neutralizing its regulatory activity. We further demonstrate that the RNA sponge mechanism is crucial for *Caulobacter* to rapidly rewire its transcriptome when switching from feast to famine conditions. Notably, mutation of *crfA* is accompanied by a severe growth defect of cells experiencing fluctuations in carbon supply, highlighting the central role of this sRNA for *Caulobacter's* oligotrophic lifestyle.

## Results

### RIL-Seq analysis reveals the Hfq-dependent RNA-RNA interactome in *Caulobacter*

To obtain a global view of the Hfq-dependent RNA-RNA interactome in *Caulobacter*, we performed RIL-seq using a 3xFLAG-Hfq expressed from the native genomic locus as bait. Cells were grown in complex PYE medium to early stationary phase (OD$_{660}$ of 1), a condition in which diverse sRNAs are expressed[18,37]. Following the original protocol[38] with minor modifications (Fig. S1a), RNA was co-immunoprecipitated with Hfq, proximal RNA termini were ligated, rRNA was depleted, and residual RNA was converted into cDNA for library preparation. As a control, cDNA libraries were generated from wild-type *Caulobacter*, producing an untagged Hfq protein, and all samples were subjected to paired-end sequencing. As expected, the number of transcripts recovered from precipitation with 3xFLAG-Hfq was in abundance over the control sample (Fig. 1a and S1b). Bioinformatic analyses revealed single fragments for which both associated sequencing reads mapped to one distinct genomic location, as well as chimeric fragments with reads mapping to independent sites on the genome[38,39]. Each chimeric fragment corresponded to a potential interaction between two transcripts, and from a total number of

484,969 chimeric cDNAs recovered with 3xFLAG-Hfq in four biological replicates, approximately 60% involved an sRNA (Fig. 1a). In total, we identified 3288 unique RNA-RNA pairs represented by at least three reads. 1541 chimeric fragments were statistically significant (FDR ≤ 0.05) and included information on the ligation point between the two recovered RNA partners (Fig. 1b). We applied ChimericFragments[39], a bioinformatics pipeline incorporating the base-pairing potential between ligated RNAs, to further select 611 high-confidence chimeras for downstream analysis (Supplementary Data 1). Among the filtered chimeras, 146 Hfq-associated interactions involved at least one sRNA (Fig. 1c, d), and approximately one-third of those sRNA-RNA pairs included SisA (CCNA_R0014), a previously uncharacterized, Hfq-associated sRNA[37].

To validate the RIL-seq approach, we assessed the potential post-transcriptional regulation by SisA with ten putative interaction partners using a GFP-based reporter system[40]. For each target, a translational fusion of the 5'-UTR and early coding sequence to the green fluorescent protein (*gfp*) was combined in *Caulobacter* with either an empty control vector or a plasmid overexpressing SisA (Fig. 1e). We observed negative regulation for eight targets (corresponding to *CCNA_02357*, *CCNA_00857*, *CCNA_00543*, *xylX*, *CCNA_03574*, *CCNA_00338*, *CCNA_03263* and *CCNA_03444*) whereas GFP production for two fusions (*CCNA_02914* and *CCNA_01807*) was only moderately affected by SisA, despite a potential interaction suggested by RIL-seq.

The predominant chimera identified in our RIL-seq dataset also included SisA; however, the sRNA was not associated with an mRNA but rather with a second sRNA, CrfA (Supplementary Data 1). CrfA is among the few sRNAs in *Caulobacter* that have been previously characterized[13,41], and, like SisA, it binds to Hfq[37]. When induced in response to carbon starvation, CrfA is required to activate the expression of more than a dozen transcripts[13]. While we were unable to recover chimeras of CrfA with putative mRNA targets, we instead, in addition to the CrfA-SisA pair, detected three less abundant interactions of CrfA with the previously uncharacterized sRNAs SisB (CCNA_R0133), SisC (CCNA_R0143) and SisD (CCNA_R0157) (Fig. 2a).

### CrfA interacts directly with the four sibling sRNAs SisA, SisB, SisC, and SisD

The four sRNAs SisA, SisB, SisC, and SisD are members of the αR8 RNA family, which is present in all Alphaproteobacteria except for the order Rickettsiales[42]. Most species carry a single αR8 RNA copy adjacent to a gene encoding a putative N-formyl glutamate amidohydrolase (FGase) protein. In *Caulobacter*, this genomic linkage is observed for *sisA*, which is located downstream of the FGase-encoding *CCNA_00780* (Fig. S3a). The additional three paralogous αR8 sRNAs are found at different locations of the genome (Fig. 2a and S3b-d). An alignment of the sequences upstream of *sisA-D* showed no conserved sequence motif (Fig. S3e), indicating that the sRNAs carry unrelated promoters with independent transcriptional control mechanisms for each sibling. In accordance, we discovered specific expression patterns for each sRNA under various growth conditions, including complex or minimal medium supplemented with different carbon sources and at different cell densities (Fig. 2e and S4a).

The SisA-D sRNAs share conserved sequence and structure traits of the αR8 RNA family: a Rho-independent terminator element, as well as an internal 10-nucleotide single-stranded stretch (5'-UUUC-CUCCCU-3') located downstream of a weak hairpin structure (Fig. 2b; S5a)[42]. Notably, this core sequence common to all αR8 RNAs is highly complementary to the second, G/A-rich stem loop of CrfA (Fig. 2c, d, S5b–e). To validate the potential interaction between the SisA-D and CrfA sRNAs, we performed structure probing experiments of CrfA in combination with Hfq and each of the four siblings. Partial digestion of in vitro–transcribed, 5' end-labelled CrfA with RNase T1 or lead (II) acetate revealed protection of residues G49-C63 upon addition of either of the four sibling sRNAs, confirming base-pairing to the

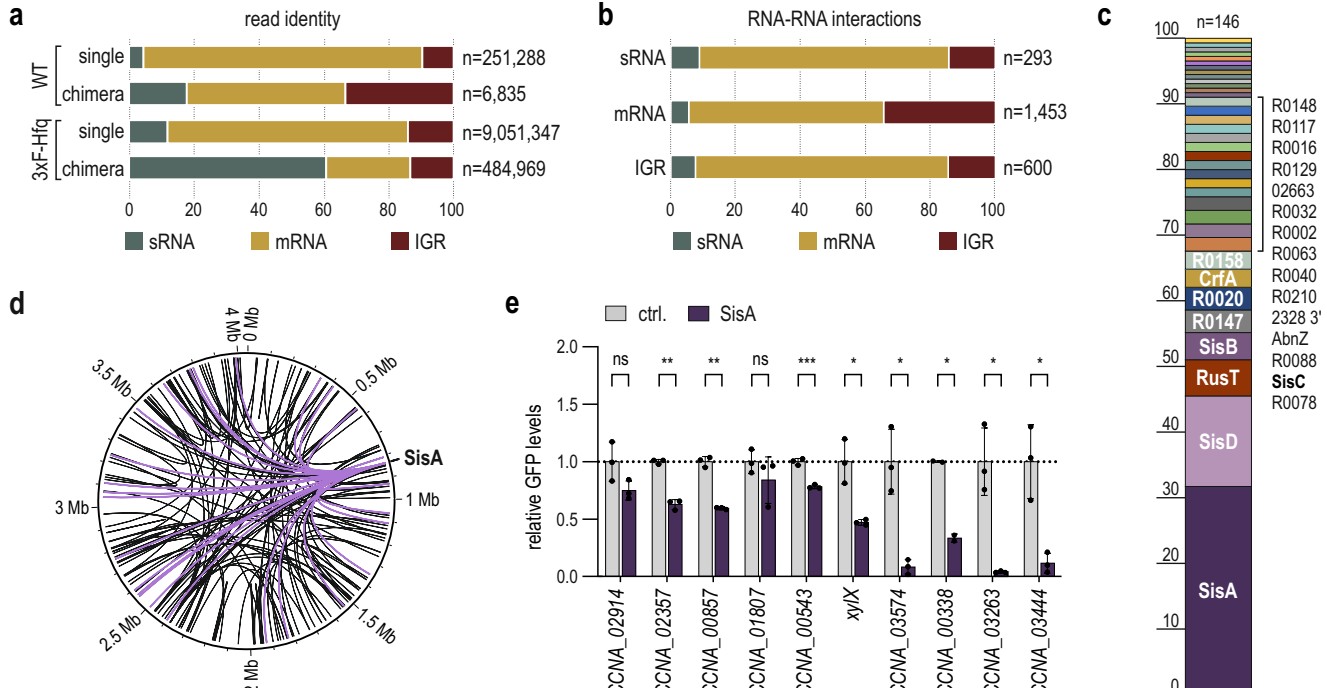

**Fig. 1 | Hfq-dependent RNA interactome in *C. crescentus* determined by RIL-seq analysis. a** Relative abundance of different RNA classes recovered from RIL-seq with *C. crescentus* wild-type (WT) or cells expressing 3xFLAG-tagged Hfq (3xF-Hfq). The total number of single or chimeric reads for each strain is indicated (*n*). **b** Distribution of interactions between different RNA classes in chimeric reads recovered with 3xFLAG-tagged Hfq. The total number of unique chimeras for each RNA class is indicated (*n*). **c** Relative distribution of interaction partners for all sRNAs. The total number of unique sRNA chimeras is indicated (*n* = 146). **d** Circos plot of the 150 most abundant RIL-seq interactions recovered with 3xFLAG-tagged Hfq. Positions on the genome are indicated. Chimeras involving SisA are highlighted in purple. **e** Validation of sRNA-mRNA interactions for SisA predicted by RIL-seq. Regulation of mRNA targets by SisA was assessed using post-transcriptional reporters encompassing the target mRNA's 5' UTR and first 20 codons fused to *gfp*.

*C. crescentus* Δ*vanAB* cells carrying the indicated reporter fusion in combination with either an empty control vector (pBVMCS-6; ctrl.) or the expression plasmid pP$_{van}$-SisA were grown overnight in the presence of vanillate to induce sRNA expression (Fig S2a). GFP expression was quantified either by fluorescence intensity measurements or by Western blot analysis of total protein samples (Fig. S2b). For the quantification of each fusion, GFP levels in the presence of the control plasmid were set to 1 and relative changes determined for cells expressing SisA. GFP regulation was calculated as mean values ± SD of three biological replicates (*n* = 3). Statistical significance was assessed using unpaired two-sided Welch's t-tests (ns for *p* > 0.05, * for *p* ≤ 0.05, ** for *p* ≤ 0.01, *** for *p* ≤ 0.001). GFP levels of a control construct did not change in the presence of SisA (Fig. S2b/c). Source data for this figure are provided as a Source Data file.

predicted site in CrfA (Fig. 2f). In a reciprocal experiment, we detected base-pairing of CrfA to SisA (residues G33-C45), covering the conserved 10-mer sequence core (Figure. S6a–c). We further demonstrated that mutation of this binding site disrupted the interaction between CrfA and SisA. Specifically, replacing residues G49-G54 with a non-complementary tetraloop sequence in CrfA (UUCG; CrfA-M2) (Fig. 2c) abolished the cleavage pattern observed for SisA in the presence of wild-type CrfA in the structure probing experiment (Fig. S6a) and interfered with RNA duplex formation of the two sRNAs on Hfq in a gel-shift assay (Fig. S6d).

We next asked how CrfA affected the expression of the αR8 RNA siblings in vivo, focusing on SisA as the most abundant representative of the family in our RIL-seq dataset (Fig. 1c). When over-expressed from a vanillate-inducible promoter, CrfA strongly decreased SisA levels (~five-fold within 10 min), whereas the sRNA was not affected by the mutated CrfA variants CrfA-M1 and CrfA-M2 in which the recognition site for SisA was disrupted (Fig. 3a, b). We further confirmed the RNA-RNA interaction in vivo through compensatory base-pair exchange experiments and expressed either CrfA wild-type or CrfA-M1 together with SisA or SisA-M1, respectively (Fig. 3c, d). Individual mutations in either RNA eliminated the downregulation of SisA by CrfA; however, this regulation was reinstated when base-pairing was restored through the combination of CrfA-M1 and SisA-M1.

For the complementary experiments, we deleted all four αR8 siblings in the *Caulobacter* chromosome (Δ*sisA-D*) to exclude any effect of native SisA-D on CrfA. As the basal levels of CrfA under non-inducing conditions are low (Fig. 2e), we also boosted expression of the sRNA by replacing the native promoter with a synthetic variant conferring constitutive, strong transcription[43]. Scoring CrfA abundance upon overexpression of SisA or SisA-M1 in comparison to a control revealed decreased full-length sRNA levels in the presence of SisA but not SisA-M1 (Fig. S7a, b). In addition, a ~65 nt-long degradation product (CrfA*) accumulated specifically upon SisA induction.

The inhibitory effects of CrfA and SisA on each other were substantially reduced in cells lacking *hfq* (Fig. 3a, b and S7a, b), indicating that the RNA chaperone facilitated RNA annealing. This result is consistent with the recovery of the interaction between the two sRNAs by Hfq RIL-seq and suggested that CrfA competes with mRNA targets for binding of SisA, in accordance with a function as a sponge RNA.

## CrfA is a sponge RNA counteracting the activity of SisA

To dissect the post-transcriptional control by CrfA and SisA, we selected *CCNA_03574* (encoding a TonB-dependent receptor protein) as a representative target. It had previously been suggested that CrfA would directly bind to the 5' UTR of the *CCNA_03574* (*CC3461*) mRNA, resulting in an activation of gene expression. Introduction of a site-specific mutation in CrfA (corresponding to variant CrfA-M1) caused a loss of the positive regulation of *CCNA_03574*, a result that had been attributed to an interruption of base-pairing between CrfA and the mRNA[13]. In contrast, our RIL-seq dataset solely recovered chimeras between the *CCNA_03574* transcript and SisA, but not CrfA (Fig. 1b; Supplementary Data 1). We thus speculated that CrfA could promote

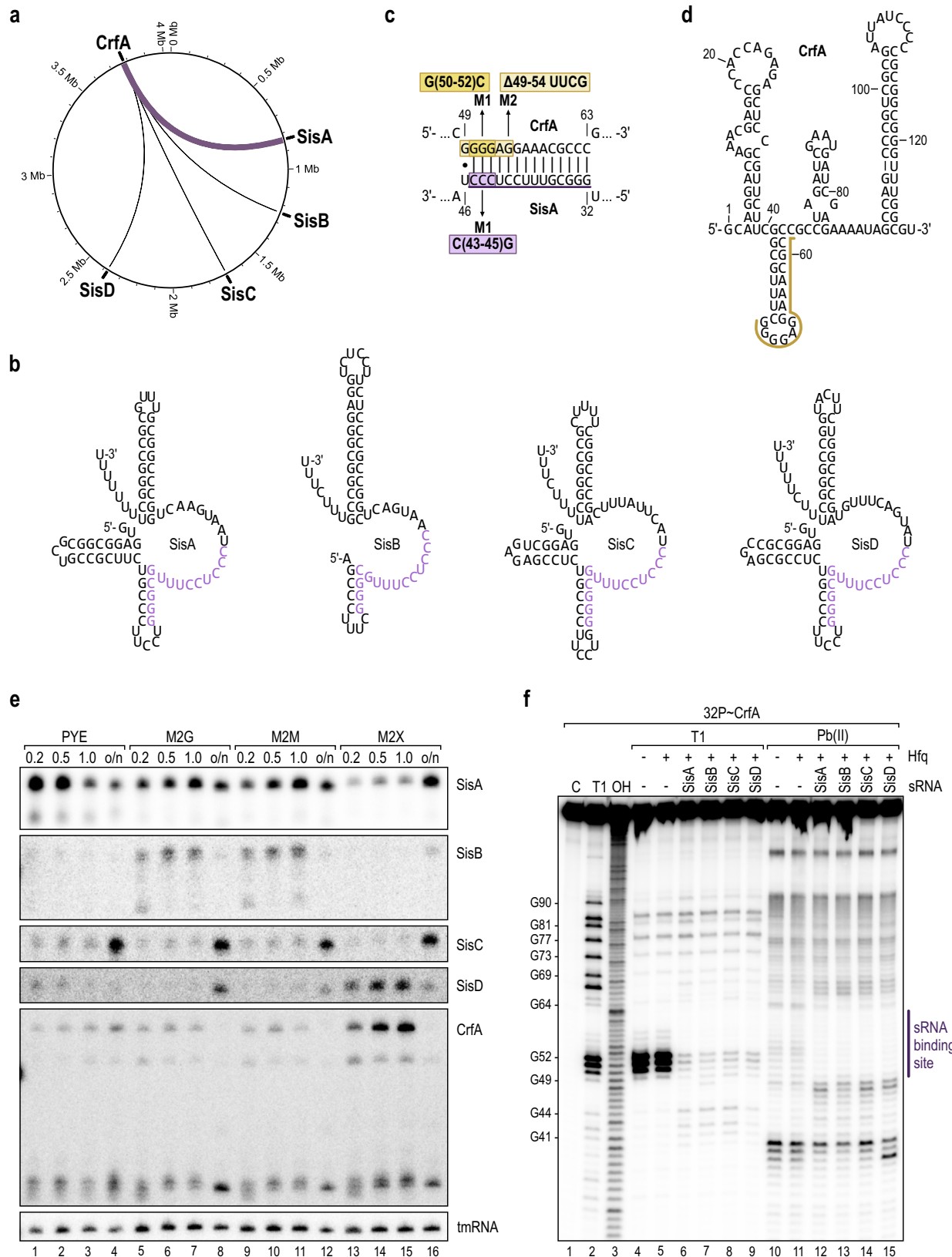

expression of *CCNA_03574* only indirectly by counteracting the negative effect of SisA on the mRNA.

To test this hypothesis, we expressed a translational *CCNA_03574::gfp* reporter in a wild-type background and tested regulation by CrfA and SisA. As expected, we observed upregulation of *CCNA_03574::gfp* in response to CrfA overexpression, but strong

downregulation by SisA (Fig. 4a; left panel). When we tested regulation of the reporter in a *Caulobacter* strain lacking all four sibling sRNAs (Δ*sisA-D*), SisA-mediated repression of *CCNA_03574::gfp* was unaffected. In contrast, CrfA over-expression failed to regulate the reporter, supporting the hypothesis that CrfA activated target gene expression by inhibiting the αR8 RNA family (Fig. 4a; right panel).

**Fig. 2 | sRNA homologues SisA, SisB, SisC and SisD interact with CrfA by direct base-pairing. a** Circos plot of CrfA interactions with the sRNA family SisA-D as recovered by RIL-seq analysis. **b** Secondary structures of SisA-D sRNAs as predicted using the RNAfold WebServer[76]. The conserved sequence stretch is highlighted in purple. **c** Predicted interaction between CrfA and SisA. Positions relative to the transcriptional start site (TSS) are indicated and nucleotide substitutions (CrfA-M1, CrfA-M2 and SisA-M1) are marked. The conserved sequence core of the SisA-D family is underlined in purple. **d** Secondary structure of CrfA as predicted using the RNAfold WebServer[76] and validated by chemical probing (Fig. 2e). The interaction site of CrfA with the SisA-D family is indicated by a yellow line. **e** Northern blot analysis of SisA-D and CrfA expression over growth in different media. *C. crescentus*

wild-type was grown in PYE or minimal medium supplemented with glucose (M2G), maltose (M2M), or xylose (M2X) as a carbon source. Total RNA samples were collected over growth at $OD_{660}$ of 0.2, 0.5, 1.0, and after overnight growth, respectively, and analysed by Northern blot. tmRNA served as a loading control. **f** In vitro structure probing of 5′-end labelled CrfA sRNA (0.4 pmol) with RNase T1 (lanes 4-9) and lead(II) acetate (lanes 10–15) in the absence and presence (5 pmol) of purified Hfq protein and SisA, SisB, SisC or SisD sRNA (4 pmol), respectively. RNase T1 and alkaline (OH) ladders of CrfA (lanes 2 and 3) were used to map cleavage fragments, and positions of mapped G-residues are marked relative to the TSS. The sRNA binding site indicated by sRNA footprints is marked with a purple bar. Source data for this figure are provided as a Source Data file.

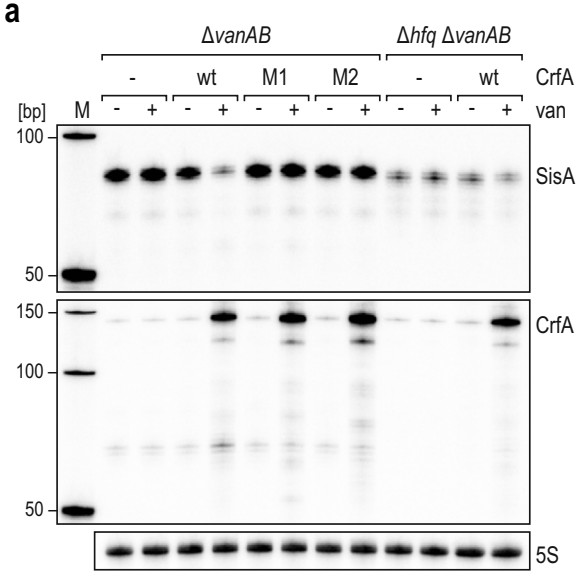

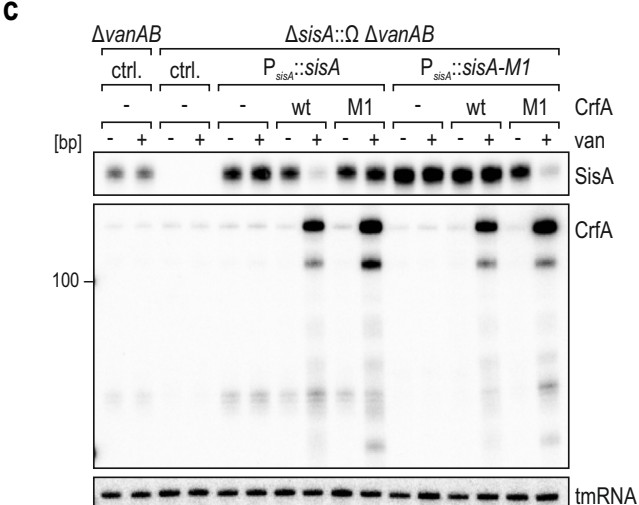

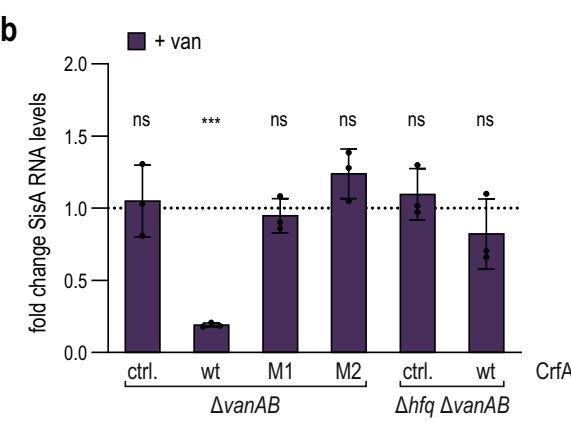

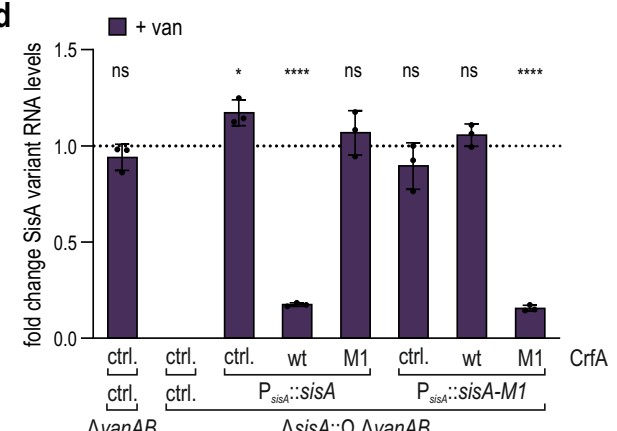

**Fig. 3 | Hfq-dependent interaction between CrfA and SisA modulates sRNA levels in vivo. a** Northern blot analysis of SisA levels in response to pulse over-expression of CrfA variants. *C. crescentus* Δ*vanAB* or Δ*hfq* Δ*vanAB* carrying either an empty control vector (pBVMCS-6; ctrl.) or plasmid pP$_{van}$-CrfA expressing CrfA wild-type (wt), CrfA-M1 (M1) or CrfA-M2 (M2), respectively, were grown to $OD_{660}$ of 0.5 in PYE. Total RNA collected before (- van) and 10 min after (+ van) addition of vanillate was analysed by Northern Blot. 5S rRNA served as loading control. **b** SisA levels after induction of CrfA variants were quantified relative to the time point before induction of the respective construct. Levels are presented as mean values ± SD for three biological replicates ($n = 3$). Statistical significance was assessed using unpaired two-sided Welch's t-tests (ns for $p > 0.05$, * for $p \le 0.05$,

** for $p \le 0.01$, *** for $p \le 0.001$, **** for $p \le 0.0001$). **c** Northern blot analysis of SisA or SisA-M1 levels in response to pulse over-expression of CrfA variants. *C. crescentus* Δ*vanAB* with a control construct or Δ*sisA*::Ω Δ*vanAB* complemented with a chromosomal construct expressing SisA or SisA-M1 from its native promoter (P$_{sisA}$::*sisA* or P$_{sisA}$::*sisA-M1*) and carrying either an empty control vector (pBVMCS-6, ctrl.) or plasmid pP$_{van}$-CrfA expressing CrfA wild-type (wt) or CrfA-M1 (M1) were grown to an $OD_{660}$ of 0.5 in PYE. Total RNA samples were collected and analysed by Northern blot as described for **a**. tmRNA served as a loading control. **d** CrfA levels were quantified as described for **b**. Source data for this figure is provided as a Source Data file.

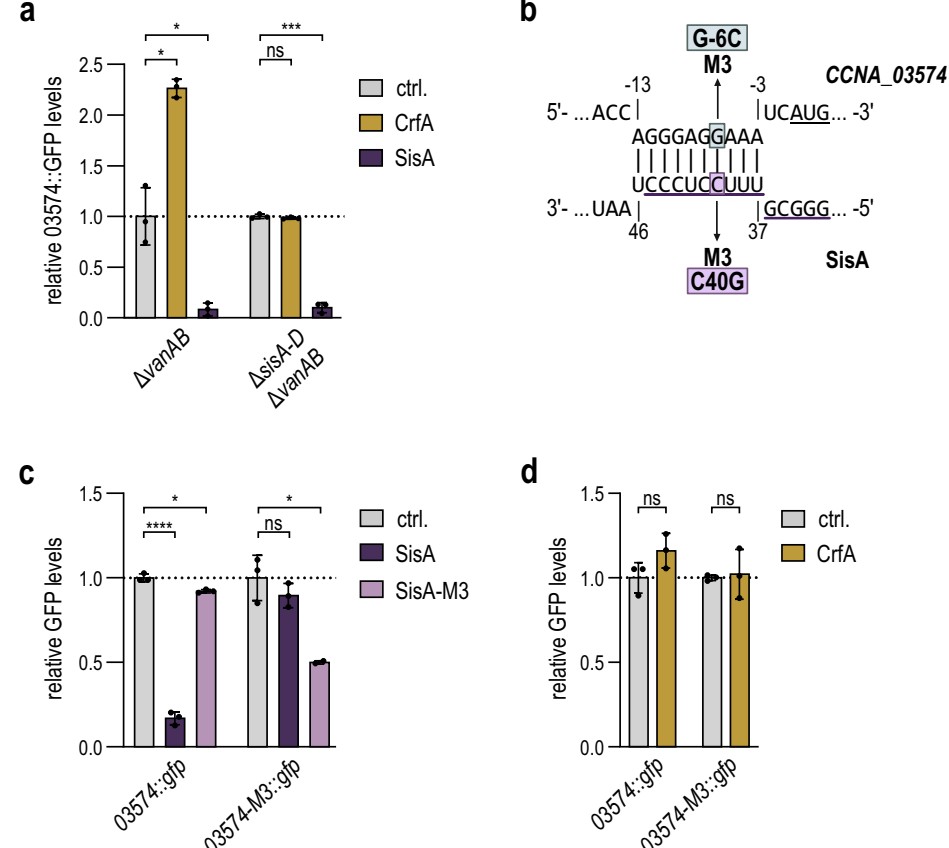

**Fig. 4 | Indirect regulation of target mRNAs through SisA-CrfA interaction. a** *C. crescentus* Δ*vanAB* or Δ*sisA-D* Δ*vanAB* carrying a chromosomal *CCNA_03574::gfp* reporter fusion in combination with either an empty control vector (pBVMCS-6; ctrl.) or the expression plasmid pP$_{van}$-CrfA or pP$_{van}$-SisA, respectively, was grown over-night in the presence of vanillate, and GFP expression was quantified by fluorescence intensity measurements. Expression of SisA and CrfA was confirmed by Northern blot analysis (Fig. S8a). GFP levels of a control construct did not change considerably in the presence of CrfA or SisA (Fig. S8c). **b** Base-pairing interaction between SisA and the *CCNA_03574* 5′UTR as predicted based on RIL-seq analysis and the IntaRNA algorithm[78]. Nucleotide positions in the sRNA are numbered relative to the TSS, in the *CCNA_03574* mRNA relative to the start codon (underlined). The conserved sequence stretch of the SisA-D family is marked in purple. Compensatory mutations SisA-M3 (C40G) and *CCNA_03574-M3* (G-6C) are indicated. **c** *E. coli* MC4100 expressing *C. crescentus* Hfq from the endogenous *hfq* locus under control of the native promoter were transformed with the *CCNA_03574::gfp* reporter fusion in combination with either an empty control vector (pKP8-35, ctrl.) or plasmid pP$_{BAD}$-SisA expressing SisA or SisA-M3. Bacteria were grown in the presence of arabinose for 6 hours, and GFP expression was quantified by fluorescence intensity measurements. Equal expression of SisA and SisA-M3 was confirmed by Northern blot analysis (Fig. S8b). GFP levels of a control construct did not change in the presence of SisA or SisA-M3 (Fig. S8d). **d** *E. coli* MC4100 expressing *C. crescentus* Hfq from the endogenous *hfq* locus under control of the native promoter were transformed with the *CCNA_03574::gfp* reporter fusion in combination with either an empty control vector (pKP8-35, ctrl.) or plasmid pP$_{BAD}$-CrfA-ribozyme expressing CrfA. Bacteria were grown in the presence of arabinose for 6 hours and GFP expression was quantified by fluorescence intensity measurements. Expression of CrfA was confirmed by Northern blot analysis (Fig. S8b). GFP levels of a control construct did not change in the presence of CrfA (Fig. S8d) **a, c, d** GFP levels were calculated as described in Fig. 1 as mean values ± SD of three biological replicates (*n* = 3), except for *CCNA_03574-M3::gfp* with SisA-M3 (*n* = 2). Statistical significance was assessed using unpaired two-sided Welch's t-tests (ns for *p* > 0.05, * for *p* ≤ 0.05, ** for *p* ≤ 0.01, *** for *p* ≤ 0.001, **** for *p* ≤ 0.0001). Source data for this figure are provided as a Source Data file.

SisA is predicted to interact with a sequence upstream of the start codon in the 5′ UTR of the *CCNA_03574* mRNA employing the 10-mer sequence motif conserved in all four sibling sRNAs (Fig. 4b). To avoid any influence of CrfA and the sibling sRNAs on the reporter, we validated the direct base-pairing between SisA and *CCNA_03574* mRNA in a heterologous system. Specifically, we over-expressed SisA in *Escherichia coli* cells producing *C. crescentus* Hfq[23], which resulted in ~six-fold repression of the *CCNA_03574::gfp* reporter. Introduction of a point mutation within the binding site in SisA-M3 (C40G) fully abrogated regulation of the reporter fusion (Fig. 4b, c), and likewise, wild-type SisA was unable to repress reporter *CCNA_03574-M3::gfp* carrying mutation G-6C. However, regulation of *CCNA_03574-M3:gfp* was restored by SisA-M3 (Fig. 4b, c), confirming the predicted interaction. Importantly, CrfA had no effect on expression of *CCNA_03574::gfp* or the M3 variant of the reporter in this background (Fig. 4d).

Together, our results indicate that SisA acts as a post-transcriptional regulator of *CCNA_03574* via direct base-pairing, while any regulatory effects on the target observed in response to CrfA expression depend on the presence of native SisA sRNA. We therefore conclude that CrfA does not directly interact with the mRNA but functions as a sponge RNA to sequester SisA, which relieves repression of its target transcript.

**Global transcriptome-wide changes by SisA and its sponge CrfA**
Our RIL-seq analysis suggested that SisA regulates multiple targets through direct base-pairing (Fig. 1d). To study if and how CrfA can interfere with SisA-mediated gene regulation at a global level, we recorded the transcriptomes of *Caulobacter* in response to over-expression of the sRNAs. To this end, we performed RNA-seq experiments of *Caulobacter* cells carrying either an empty control vector,

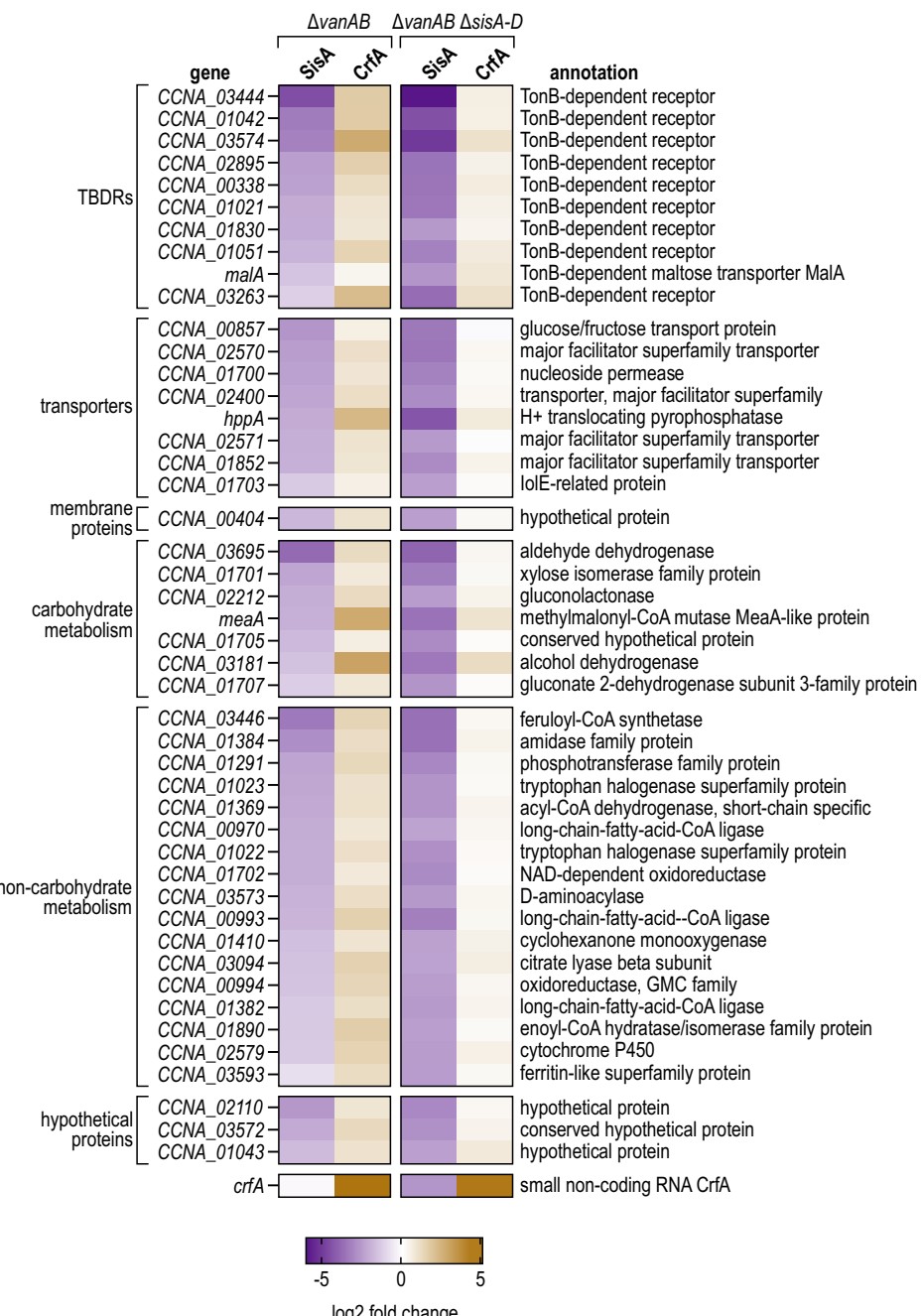

**Fig. 5 | Differential gene regulation in response to pulse-overexpression of SisA and CrfA.** *C. crescentus* Δ*vanAB* or Δ*sisA-D* Δ*vanAB* carrying either an empty control vector (pBVMCS-6) or the expression plasmid pP$_{van}$-SisA or pP$_{van}$-CrfA, respectively, were grown in M2G to an OD$_{660}$ of 0.3. Total RNA was extracted from cells before and 15 minutes after the addition of vanillate, and used to prepare cDNA libraries for high-throughput sequencing. Gene expression changes in response to pulse-induction of SisA or CrfA, respectively, are plotted as a heatmap. Expression values are listed in Supplementary Data 2. Genes with fold-repression ≥ 5 (and FDR corrected *p*-value of ≤ 0.05, see Methods for details) in response to pulse-induction of SisA in either Δ*vanAB* or Δ*sisA-D* Δ*vanAB* are shown, with fold changes (regardless of the FDR corrected p-value) in response to pulse-induction of CrfA in the same genetic background. Source data for this figure are provided as a Source Data file.

pP$_{van}$-CrfA, or pP$_{van}$-SisA, which we cultivated in minimal M2 medium supplemented with 0.2% glucose (M2G) to mid-exponential phase (OD$_{660}$ of 0.3). To specifically address the sponge activity of CrfA, we also performed a corresponding transcriptome analysis in *Caulobacter* Δ*sisA-D* cells. RNA was prepared from samples collected prior to as well as 15 min after addition of the inducer vanillate, and CrfA or SisA overexpression was verified by Northern blot analysis (Fig. S9). Expression profiles for the individual sample sets were determined by RNA-seq analysis, which revealed differential expression of 47 transcripts (FDR p-value ≤ 0.05) (Supplementary Data 2). In line with our hypothesis and the previous characterization of the sRNA[13], CrfA over-expression resulted in upregulation of numerous targets, including *CCNA_03574*. The same set of transcripts was also differentially expressed upon SisA overexpression (≥10 reads; ≥ 5-fold change; FDR *p*-value ≤ 0.05), however, displaying opposite regulation. In contrast, in *Caulobacter* Δ*sisA-D*, CrfA failed to activate these targets, whereas SisA also functioned as a negative regulator in this background (Fig. 5).

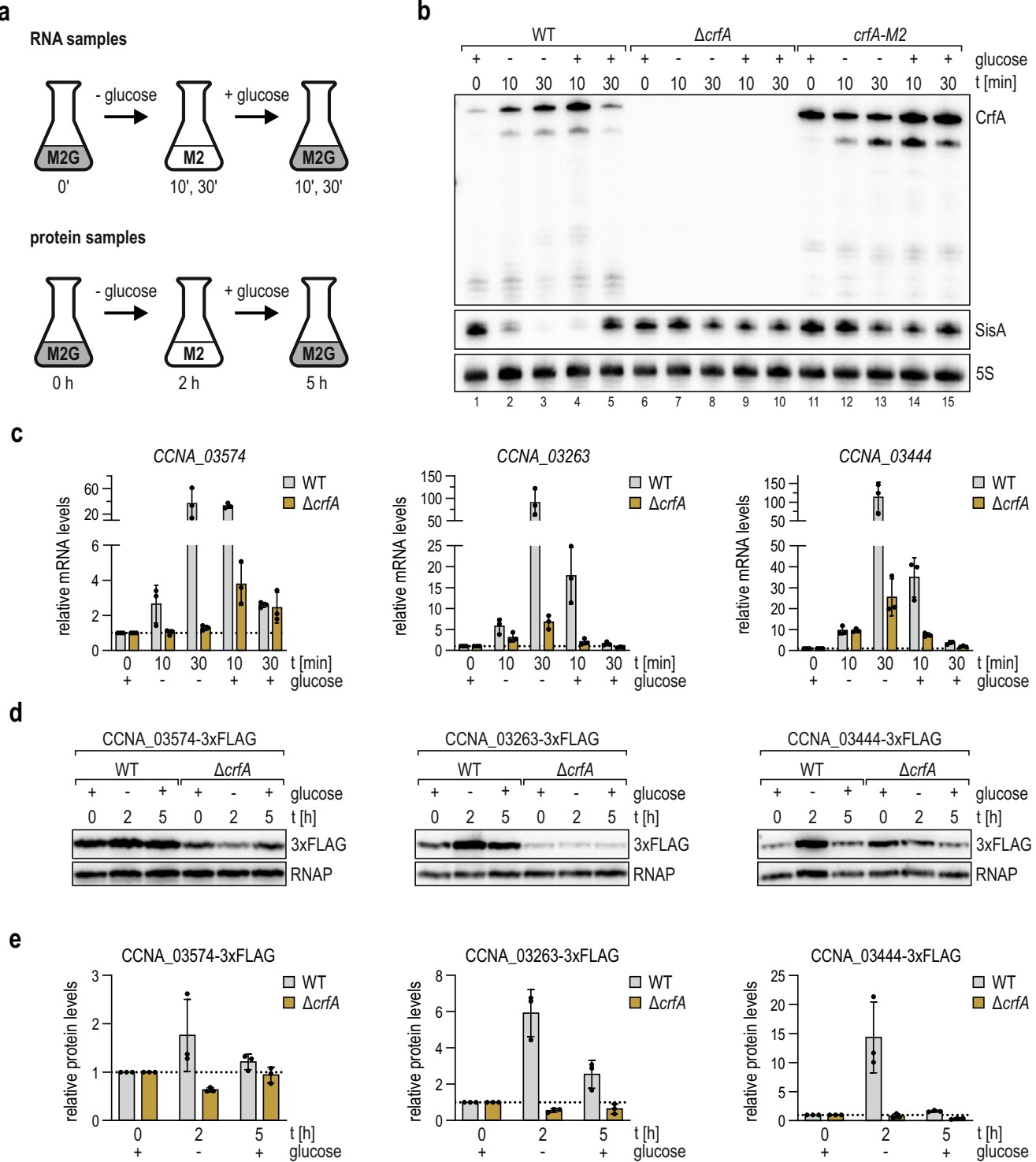

**Fig. 6 | CrfA-dependent modulation of SisA and its target mRNAs during carbon starvation. a** Schematic of carbon starvation experiments and respective sample collection time points. *C. crescentus* wild-type, Δ*crfA* or *crfA-M2*, respectively, were grown to $OD_{660}$ of 0.2 in minimal medium supplemented with glucose (0.2%; M2G), washed twice and resuspended in minimal medium without carbon source (M2). Total RNA samples were collected after 10 min and 30 min, respectively. Total protein samples were collected after 2 hours. After the last sample collection time point (30 min for RNA, 2 h for protein, respectively), glucose was added again (final concentration 0.2%) and total RNA sampled after 10 min and 30 min, respectively, while total protein samples were collected after 5 h. **b** Expression of CrfA and SisA during starvation and recovery was analysed on Northern blots using RNA samples prepared from *C. crescentus* wild-type, Δ*crfA*

and *crfA-M2* grown as described in (**a**). **c** SisA target gene mRNA levels of *CCNA_03574*, *CCNA_03263* and *CCNA_03444*, encoding TonB-dependent receptors, during starvation were quantified by RT-qPCR on total RNA collected from *C. crescentus* wild-type or Δ*crfA* grown as described in (**a**); levels are presented as mean values ± SD of three biological replicates (*n* = 3). **d** Expression of CCNA_03574-3xFLAG, CCNA_03263-3xFLAG and CCNA_03444-3xFLAG during starvation was analysed by Western blot using total protein samples collected from *C. crescentus* wild-type or Δ*crfA* grown as described in (**a**). **e** Protein levels were quantified relative to the protein abundance prior to starvation in the wild-type and are presented as mean values ± SD of three biological replicates (*n* = 3). Source data for this figure are provided as a Source Data file.

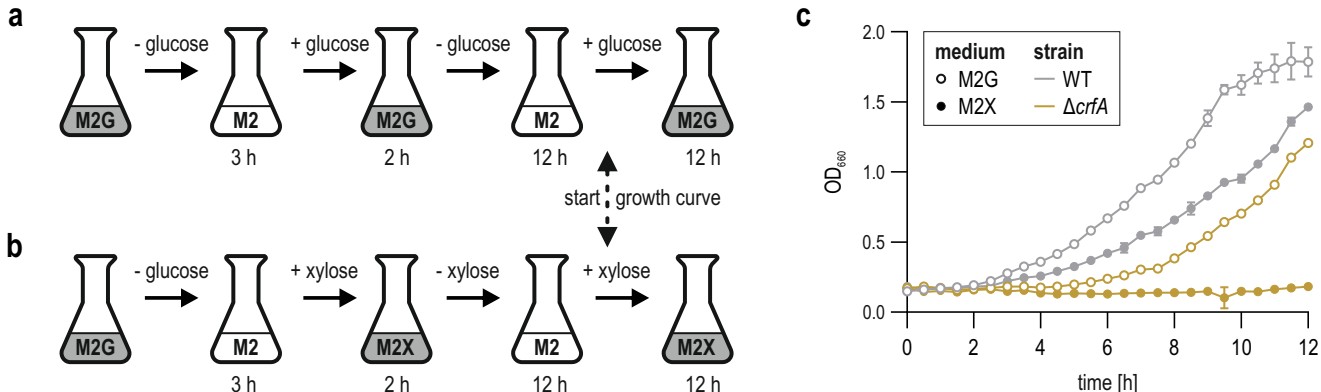

**Fig. 7 | Absence of CrfA negatively impacts recovery after carbon starvation.**
**a**, **b** *C. crescentus* wild-type and Δ*crfA* were grown to an $OD_{660}$ of 0.2 in M2G, subjected to starvation in M2 without a carbon source for 3 hours, and recovered by the addition of 0.2% glucose (**a**) or 0.2% xylose (**b**). After 2 hours of growth, cells were starved again in M2 without a carbon source for 12 hours and then again recovered with 0.2% glucose (**a**) or 0.2% xylose (**b**), respectively, for 12 hours.

**c** Growth during the second recovery with glucose (open symbols) or xylose (closed symbols) was monitored over 12 hours. Mean optical density of wild-type (grey) and Δ*crfA* (gold), respectively, was calculated from three biological replicates ($n = 3$), and error bars show the standard deviation. The source data underlying this figure are provided as a Source Data file.

To corroborate our findings, we validated post-transcriptional regulation of four additional representative targets by CrfA and SisA, respectively, in a strain lacking all four sibling RNA genes using post-transcriptional reporters (Fig. S10). As observed for *CCNA_03574* (Fig. 4b, c), fusions of *CCNA_03263*, *hppA*, *CCNA_03444*, and *CCNA_00338* to *gfp* were downregulated by SisA but not CrfA (Fig. S10a, b). In all cases, we were able to predict base-pairing interactions between SisA and the target candidates involving the conserved SisA seed (Fig. S10c–f). Collectively, our results reveal a role for SisA as a post-transcriptional regulator of a broad network of mRNA transcripts, while CrfA only indirectly contributes to target regulation by counteracting the αR8 sRNA.

## CrfA and SisA have opposing roles in response to carbon starvation

Expression of CrfA is induced by carbon depletion[13]. The extensive regulatory interplay between CrfA and SisA observed upon over-expression of the sRNAs prompted us to determine whether the sponging activity of CrfA helped *Caulobacter* adapt to fluctuations in carbon supplies that cells may encounter in their natural nutrient-poor habitat. In analogy to the above experiments, we focused on SisA as it was the most abundant sibling in our experiments (Fig. S4a). To induce starvation, *C. crescentus* was grown in M2G to exponential phase ($OD_{660}$ of 0.2), washed twice, and subsequently resuspended in M2 without any carbon source. Cultivation was continued for 30 min when carbon deprivation was relieved by addition of glucose to the medium (Fig. 6a; top panel). In *Caulobacter* wild-type cells, CrfA was highly induced (~six-fold) within 10 min in M2 and decreased again after supplementing glucose. Inversely, SisA abundance strongly decreased (~50-fold within 30 min) in response to carbon depletion and returned to the initial level within 30 min of recovery in M2G (Fig. 6b). In the absence of CrfA (Δ*crfA*), or in cells carrying a chromosomal mutation in *crfA* interrupting base-pairing with SisA (*crfA-M2*, Fig. 2c, Fig. 3a), SisA levels changed only moderately throughout the phases of starvation and recovery (Fig. 6b). Of note, basal levels of CrfA-M2 were elevated compared to wild-type CrfA in M2G (compare lanes 1 and 11), suggesting that base-pairing interactions with SisA, and possibly its paralogs SisB-D, contributed to the reduced expression of CrfA under carbon-rich conditions.

These results indicated that CrfA was the major factor controlling SisA expression during starvation, and we were curious how this additional level of post-transcriptional regulation affected SisA target genes. To address this question, we monitored mRNA and protein levels of five SisA target genes in response to short-term and long-term glucose depletion and recovery in the presence or absence of CrfA (Fig. 6a). As determined by RT-qPCR, the expression patterns of five representative target mRNAs (*CCNA_03574*, *CCNA_03263*, *CCNA_03444*, *hppA*, and *CCNA_00338*) in wild-type *Caulobacter* followed the dynamics of CrfA, with strongly increasing expression in response to carbon depletion and concomitant down-regulation to basal levels upon readdition of glucose (Fig. 6c, Fig. S11a). In cells lacking *crfA*, the basal levels of all five transcripts were decreased compared to the wild-type strain before starvation, and mRNA abundance either barely changed after carbon depletion (*CCNA_03574*, *hppA*, *CCNA_00338*), or induction during starvation was strongly reduced (*CCNA_03263*, *CCNA_03444*).

To assess whether this regulatory pattern was also reflected at the protein level, we inserted a 3xFLAG-tag (3xF) epitope sequence upstream of the annotated stop codon of four TBDRs (*CCNA_03574*, *CCNA_00338*, *CCNA_03263*, and *CCNA_03444*) and a H+ translocating pyrophosphatase (*hppA*). We extended the periods of starvation and recovery, respectively, to follow protein dynamics and collected total protein samples prior to as well as after 2 h of starvation and 5 h of recovery (Fig. 6a; lower panel). For wild-type cells, we observed an increase of the four TBDRs after 2 h of starvation, followed by a decrease to nearly basal levels 5 h after the addition of glucose for all target candidates except HppA-3xF, for which expression remained high (Fig. 6d, e, Fig. S11c). In contrast, in the absence of *crfA*, protein levels for all four targets barely changed despite the fluctuating carbon supply, and, except for CCNA_03444-3xF, basal levels were decreased compared to the wild-type (Fig. 6d, e).

## *Caulobacter* depends on CrfA to adapt to fluctuating environments

The extensive regulon of SisA recovered by RIL-seq and pulse expression suggested comprehensive, CrfA-dependent remodelling of the *Caulobacter* gene expression profile in response to starvation. We therefore asked whether the sponge activity was important for the bacterium's stress resilience. Specifically, we mimicked fluctuating nutrient availability by two consecutive rounds of starvation (3 h and 12 hours, respectively) interrupted by a short recovery phase (2 h; see Fig. 7a, b) and monitored cell outgrowth in carbon-supplemented medium (Fig. 7c). When carbon repletion was realized through the addition of glucose (open symbols), wild-type cells quickly adapted and reached stationary phase ($OD_{660} \geq 1.6$) within ~9.5 hours. In contrast, *crfA* mutant cells showed a significantly delayed response, but eventually resumed growth.

We also addressed whether the choice of carbon source provided after starvation would impact the recovery phase. In its natural freshwater habitat, *Caulobacter* may feed on plant-derived biopolymers, with glucose and xylose being the major monomeric sugars released during the degradation of abundant lignocellulose[44]. Replenishing the medium with xylose instead of glucose modestly reduced the growth rate of the wild-type strain (Fig. 7c; closed symbols). Importantly, the switch to another sugar after starvation resulted in an almost complete inhibition of growth of Δ*crfA* cells, suggesting that the CrfA-mediated reprogramming of the *Caulobacter* transcriptome is necessary to reorient the cellular metabolism to the different carbon source.

## Discussion

Our understanding of the key functions fulfilled by bacterial sRNAs in regulatory circuits and stress responses has grown tremendously over the past two decades. While pioneering work established the enterobacteria *E. coli* and *Salmonella* Typhimurium as model organisms of microbial RNA biology, the advent of high-throughput sequencing technologies has promoted the comprehensive identification of sRNAs in diverse microbial species[45]. However, the functional characterization of individual sRNAs has remained a major bottleneck. Recently, experimental methodologies based on RNA proximity ligation have facilitated the annotation of global RNA interactome maps and helped elucidate the functional context in which distinct sRNAs are embedded[46]. Here, we report RIL-seq mapping of the Hfq-dependent RNA-RNA network in *C. crescentus*. Based on our dataset, we identify the sRNA CrfA as an antagonist of a family of sRNAs, SisA-D, and demonstrate a role for the RNA sponge in remodelling the *Caulobacter* transcriptome in response to fluctuating carbon availability.

Our study substantially expands the RNA-RNA network within *Caulobacter*. The fraction of chimeric transcripts within the total population of sequences (-5%) obtained by RIL-seq was within the same range of what has been observed for other bacteria[28,29,47,48], confirming the successful implementation of the protocol in *Caulobacter*. The stringent processing of raw reads through the ChimericFragments pipeline[39] resulted in a set of ~600 potential RNA-RNA interactions with high confidence (Supplementary Data 1). Using translational reporter fusions, eight out of ten selected putative targets of SisA were confirmed to be regulated by the sRNA at the post-transcriptional level (Fig. 1e). Likewise, the interaction between CrfA and the SisA-D family retrieved from the RIL-seq dataset was verified by additional experimental approaches (Fig. 2f; 3a, b). Collectively, these results are indicative of a high degree of reliability of our RNA interactome dataset.

In addition to sRNA-mRNA or sRNA-sRNA pairs, we also recovered numerous potential interactions between two protein-coding transcripts (Fig. 1b). In part, these chimeras may not include the full-length mRNA but rather stable processing intermediates. This class of transcripts has more recently emerged as an abundant sRNA subtype that is often missed in current annotations. For example, RIL-seq experiments in *E. coli*, *K. pneumoniae* and *V. cholerae* identified the sRNAs GadF, DinR and FlaX, respectively, which are all derived from mRNA 3′ ends[36,48,49]. It remains to be determined how many intact mRNAs are involved in the obtained RNA-RNA pairs, and what the functional consequences of an mRNA-mRNA interaction might be. Indeed, pairing between two full-length mRNAs can fulfil a regulatory role, as has been exemplified by the stabilization of the *prsA2* mRNA through base-pairing with the *hly* mRNA in *Listeria monocytogenes*[50], the *vigR* mRNA-mediated protection of *folD* and *isaA* mRNAs from endonucleolytic cleavage in *Staphylococcus aureus*[51] or the modulation of *gbpC* mRNA turnover through direct duplex formation with *irvA* mRNA in *Streptococcus mutans*[52].

Another interesting aspect of Hfq-mediated regulation in *Caulobacter* is the specificity of the RNA chaperone for its ligands. In general, *Caulobacter* Hfq shows reduced affinities and higher selectivity toward substrate binding when compared to the *E. coli* homologue[23]. In addition, not all sRNAs recovered with Hfq in *Caulobacter* carry the typical signature binding site associated with standard sRNAs in enterobacterial species, namely an AU-rich sequence stretch upstream of a Rho-independent transcriptional terminator consisting of a strong stem-loop structure followed by a U-run[37,53]. Nevertheless, we observe a clear overrepresentation of sRNAs in read 2 rather than read 1 of chimeric sequences (73 vs. 27%; Fig. S1c), as it has been detected for RIL-seq studies in other species[36,47,48]. This tendency reflects a mode for sRNA binding in which the 3′ end anchors the transcript to Hfq and is not accessible for ligation. Less is known about the conformation of mRNAs on the RNA chaperone, or the orientation of two interacting sRNAs on *Caulobacter* Hfq. The latter scenario is relevant for the Hfq-dependent sponge activity of CrfA on the SisA-D family (Fig. 2). With CrfA being almost exclusively detected in read 1 of the chimeras (Supplementary Data 1), i.e., the "mRNA position", it will be interesting to study how CrfA competes with target mRNA transcripts for SisA-D base-pairing, and how species-specific characteristics of the Hfq chaperone contribute to this regulatory process.

In some cases, a single microbe encodes two or more highly similar sibling sRNAs that likely originate from gene duplication events[54]. The selective pressure behind the retention of multiple, seemingly equivalent regulators is not understood in all instances. A high degree of sequence conservation among siblings suggests comparable base-pairing capacities of the sRNAs and implies overlapping functions in gene regulation. Well-studied examples for siblings that act redundantly are OmrA and OmrB, two sRNAs encoded in tandem in the intergenic region between *aas* and *galR* on the genome of *E. coli* and related enterobacteria[55]. Both Hfq-dependent RNAs are induced via the EnvZ-OmpR two-component system under high osmolarity conditions and share an almost identical, conserved sequence at the 5′ end. Base-pairing of this common site with several mRNAs represses the synthesis of outer membrane transporters, siderophore receptors, and transcriptional regulators, resulting in substantial remodelling of the membrane[56–58]. Likewise, depending on the species, up to five members of the Qrr sRNA family conserved in *Vibrio spp.* regulate a mostly overlapping set of target mRNAs to fine-tune the cell's intrinsic quorum-sensing circuit, suggesting that a higher copy number of the RNA is required to comprehensively control the system[59,60]. Although the sibling sRNAs are encoded dispersed across the *Vibrio* chromosome, all Qrrs are co-regulated, and their expression is restricted to conditions in which autoinducer concentrations are low[60,61]. The siblings SisA-D in *C. crescentus*, in contrast, exhibit distinct expression patterns with no conserved motifs within their respective promoter sequences (Fig. S3e). SisA is the most abundant family representative exhibiting nearly constitutive expression under the conditions tested (Fig. 2e and S4a). Our transcriptome dataset revealed that, when overexpressed, SisA represses a large set of targets likely facilitated via base-pairing through a 10-mer motif with an anti-Shine-Dalgarno signature shared by all four sRNAs. It is therefore possible that SisA-D has the potential to control an overlapping set of targets. However, due to their differential transcriptional control, each sibling may also regulate individual transcripts present only under the respective condition.

Based on phylogenetic analyses, a combination of sequence duplication with vertical and lateral gene transfer accounts for the distribution of the widely conserved αR8 sRNA family within the Alphaproteobacteria, represented in *C. crescentus* by SisA, SisB, SisC, and SisD[42]. The first and up to now only characterized sRNA within the αR8 family is MmgR, an Hfq-associated regulator of the root symbiont *Sinorhizobium meliloti*[62,63]. While a direct base-pairing interaction of MmgR has not been identified, *S. meliloti* expressing a variant of the sRNA lacking the characteristic 10-mer sequence displayed over-accumulation of the carbon storage polymer polyhydroxybutyrate (PHB) synthesized under growth-limiting conditions[64]. MmgR is the sole αR8 sRNA in *S. meliloti*, but many genera, including the

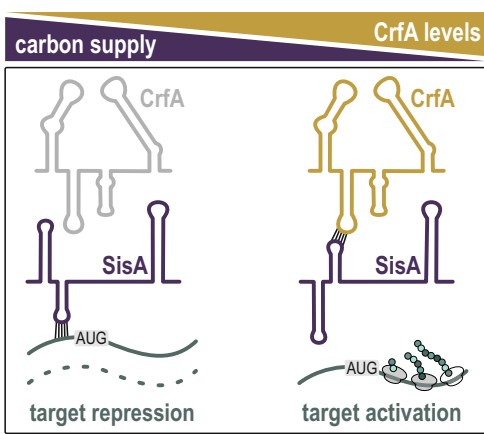

**Fig. 8 | CrfA is a sponge RNA antagonizing SisA activity.** Under nutrient-replete conditions, the SisA small RNA negatively regulates the expression of its target genes by directly base-pairing with mRNAs. Upon carbon depletion, the abundance of CrfA increases. Through direct base-pairing, CrfA binds to and sequesters SisA as a sponge RNA. This allows mRNA de-repression and ultimately contributes to adaptation to nutrient-poor conditions.

*Caulobacterales* contain multiple copies of the sRNAs, some of which are encoded adjacent to each other or at independent genomic loci[42]. Future studies will address whether all or only a subset of the αR8 sRNAs are relevant for carbon metabolism in their respective species, and whether the sibling sRNAs act redundantly or rather independently on their cognate target transcripts.

The regulatory activity of the previously mentioned Qrr sRNA family of marine *Vibrio* species is antagonized by another sRNA, QrrX, which functions as an RNA sponge[28]. QrrX expression refines the transition between individual to community behaviour in *Vibrio* by accelerating the decay of the Qrr siblings when no longer needed.

Likewise, SisA-D sRNAs are counteracted by an RNA sponge, CrfA (Figs. 2, 3 and 8), which masks the sRNAs' interaction site with mRNA targets. Originally, CrfA was mistaken as an mRNA-pairing sRNA that activated expression of its targets[13]. For *CCNA_03574*, a predicted interaction within the very 5' end of the target transcript was indeed sensitive to a mutation in CrfA; however, we now show that this exchange does not interfere with CrfA recognition of the mRNA but rather alleviates base-pairing of CrfA within the conserved internal sequence element common to all four siblings (Fig. 2b, c). Although CrfA is only expressed at low basal levels under carbon-replete conditions, the abundance of SisA-D increases in the absence of *crfA* (Fig. S4b). When CrfA is induced during carbon starvation, we observed a strong reduction in SisA-D expression levels, suggesting that the sponge triggers decay of the siblings (Figs. 3 and 6). Which RNases facilitate turnover of the sRNAs, and whether CrfA acts catalytically or is possibly co-degraded with its interaction partners remains to be determined. In response to carbon starvation, SisA appears to be the relevant sRNA to be inactivated by CrfA as it is more abundant compared to the other family members and remains constitutively expressed in the absence of the sponge (Fig. 6b). We detected robust upregulation of selected targets once SisA was depleted through CrfA, but barely any regulation in a *crfA* mutant strain (Fig. 6c−e and S11). The functional categorization of the SisA target set revealed a strong representation of TBDRs and other membrane transporters, as well as various metabolic factors (Fig. 5). A restructuring of the cell envelope and a focus on the degradation of plant-derived carbon sources can be interpreted as mechanisms to prepare cells for the uptake and utilization of certain carbohydrates.

For several reasons, the reorganization of the transcriptome during starvation via an RNA-centric system is favourable for the bacterium. First, the CrfA sponge provides a mechanism for the rapid turnover of

SisA, an sRNA with a long half-life of more than 30 min under carbon-replete conditions[65]. Second, using an RNA regulator rather than a protein is more energy-efficient, conserving resources when nutrients are limiting. Third, the interplay between the two sRNAs enables positive as well as negative regulation of their target set, allowing for rapid adaptation to fluctuating environments (Fig. 8). The central role of CrfA under this stress is evidenced by the severe growth defect observed when the sponge is absent under erratic nutrient supply (Fig. 7c). Of note, CrfA does not respond to phosphate or nitrogen starvation[13], but its transcriptional regulation as well as its potential activity under alternative stress conditions remain to be determined.

In sum, our RIL-seq study has uncovered a wealth of Hfq-dependent RNA-RNA interactions in *C. crescentus*. The key role of the interplay between CrfA and its sRNA targets for rewiring the gene expression profile in response to carbon starvation underscores the importance of RNA-mediated regulation in this species. In this regard, we consider our interactome dataset a substantial resource for future studies and have made it easily accessible via a web browser: https://caulobacter-interactome.uni-jena.de/.

## Methods

### DNA oligonucleotides

Sequences of all oligonucleotides employed in this study are listed in Supplementary Table S1.

### Construction of plasmids

All plasmids used in this study are summarized in Supplementary Table S2. Plasmids for allelic replacements were constructed by fusing flanking fragments of genes *crfA* (f1: KFO-1308/KFO-1309; fΩ: KFO-0636/0637; f2: KFO-1310/1311); *sisA* (f1: KFO-0169/KFO-1196; fΩ: KFO-0636/0637; f2: KFO-1197/172); *sisB* (f1: KFO-1202/KFO-0572; f2: KFO-0573/1203); *sisC* (f1: KFO-1236/KFO-1237; f2: KFO-1238/1239); *sisD* (f1: KFO-0177/KFO-0178; f2: KFO-0179/0180) with linearized plasmid pNPTS138 (KFO-0059/KFO-0060) at the multiple cloning site using Gibson assembly according to the manufacturer's recommendation (NEB, #E2611). For allelic replacement of *crfA* with a variant, the Ω-cassette in plasmid pKF667-1 was replaced with the *crfA* locus (pNPTS-*crfA*; pKF767-2) by fusing the PCR amplified backbone (KFO-1721/KFO-1694 on pKF667-1) with the *crfA* fragment (PCR amplified from gDNA using KFO-1695/KFO-1696) by Gibson assembly.

Plasmids for C-terminal 3XFLAG tagging were constructed by fusing flanking fragments of genes *CCNA_03574* (f1: KFO-2530/KFO-2531; f2: KFO-2532/KFO-2533); *CCNA_03263* (f1: KFO-2542/KFO-2543; f2: KFO-2544/KFO-2545); *hppA* (f1: KFO-2566/KFO-2567; f2: KFO-2568/KFO-2569); *CCNA_03444* (f1: KFO-2562/KFO-2563; f2: KFO-2564/KFO-2565); *CCNA_00338* (f1: KFO-2534/KFO-2535; f2: KFO-2536/KFO-2537) with linearized plasmid pNPTS138 (KFO-0059/KFO-0060) at the multiple cloning site using Gibson assembly according to the manufacturer's recommendation (NEB, #E2611).

For plasmids pP$_{van}$-SisA and pP$_{van}$-CrfA, the sRNA fragment (PCR amplified from *C. crescentus* gDNA using KFO-0141/KFO-0145 and KFO-0668/KFO-0712, respectively) was ligated by Gibson assembly to the pBVMCS-6 backbone ([66]; amplified with KFO-0144/KFO-0198). For expression of SisA from the inducible P$_{BAD}$ promoter (pKF1014-1) in *E. coli*, the *sisA* fragment was PCR amplified (KFO-2643/KFO-2644 on pKF384-1) and ligated to the pBAD5A backbone (pKP8-35[67] amplified with KPO-0196/KPO-0411) by Gibson assembly. To express CrfA in *E. coli*, the *crfA* fragment was amplified (KFO-712/KFO-668), digested with XbaI, and first ligated into the equally cleaved pZE12-luc backbone ([68]; amplified with KFO-1544/PLlacOD) to create pKF668-4. To improve transcription termination of CrfA, the self-cleaving HDV-ribozyme[69] sequence was added to the 3' end of the sRNA (pKF996-1) by ligating the ribozyme (fragment generated by self-annealing of KFO-2955/KFO-2956) with the pKF668-4 plasmid (linearized by

PCR with KFO-2953/KFO-2954) by Gibson assembly. The CrfA-ribozyme insert was amplified (KFO-2645/KFO-3328 on pKF996-1) and ligated to the pBAD5A backbone (pKP8-35[67]; linearized with KPO-0196/KPO-0411) by Gibson assembly to create pP$_{BAD}$-CrfA-ribozyme (pKF1064-1).

Constitutive expression of CrfA was achieved by replacing the inducible P$_{van}$ promoter with the constitutive J23119 promoter[43]. To this end, pKF480-7 was PCR amplified with KFO-0635/KFO-1780, the PCR product digested with DpnI, and, following purification, self-ligated. The P$_{const}$-crfA fragment was then PCR amplified (KFO-0684/KFO-1781) and cloned into XbaI and NdeI-restricted pXGFPC-4 to create pKF777-2 (pP$_{const}$-crfA).

To complement ΔsisA::Ω ΔvanAB with SisA, the sisA fragment, including its promoter region (100 bp upstream the TSS) was amplified by PCR (KFO-1841/KFO-1842 on C. crescentus gDNA) and cloned into NheI/NdeI restricted pXGFPC-4 to create pKF778-3 (pP$_{sisA}$-sisA). For post-transcriptional gfp reporter fusions under control of the constitutive rsaA promoter, the 5′ UTR and first 20 codons of each target gene (insert amplified from gDNA, see Table S2 for details) were cloned into EcoRI and KpnI-restricted pKF385-2 as described previously[40]. The reporter plasmids were designed to integrate at the rsaA locus in the C. crescentus genome. For expression of the post-transcriptional GFP reporter CCNA_03574::gfp under control of the constitutive P$_{LTetO1}$ promoter (pKF682-1), the 5′ UTR and first 20 codons of CCNA_03574 (amplified from gDNA using KFO-1371/KFO-1372) were fused with the linearized pXG10 backbone (amplified with KPO-7614/KPO-1702) by Gibson assembly.

Nucleotide mutations were introduced by PCR amplification of the template plasmid, digestion of the PCR product with DpnI and self-ligation after purification. Plasmid pKF480-7 served as template for PCR amplification with oligo pairs KFO-2497/KFO-2498 (pP$_{van}$-CrfA-M1; pKF882-1) and KFO-1306/KFO-1307 (pP$_{van}$-CrfA-M2; pKF666-1); pKF348-1 served as template for PCR amplification with oligo pair KFO-2495/KFO-2496 (pP$_{van}$-SisA-M1; pKF881-1); pKF1014-1 served as template for PCR amplification with oligo pair KFO-2983/KFO-2984 (pP$_{BAD}$-SisA-M3; pKF1022-1); pKF682-1 served as template for PCR amplification with oligo pair KFO-2951/KFO-2952 (pP$_{LtetO1}$-CCNA_03574-M3::gfp; pKF999-1); pKF767-2 served as template for PCR amplification with oligo pair KFO-1306/KFO-1307 (pNPTS-crfA-M2; pKF769-1); pKF778-3 served as template for PCR amplification with oligo pair KFO-2495/KFO-2496 (pP$_{sisA}$-sisA-M1; pKF897-1).

## Construction of bacterial strains and growth conditions

Bacterial strains used in this study are listed in Supplementary Table S3. Caulobacter crescentus strain NA1000 (KFS-006) is referred to as the wild-type strain and was used for mutant construction. Genomic deletions and insertions in C. crescentus were obtained using a two-step recombination[70]. Chromosomal mutations were transferred by transduction with phage Cr30 following standard protocols[71].

C. crescentus was cultivated aerobically at 30 °C in either complex PYE medium, or in minimal M2 salts containing 0.2% glucose, 0.3% maltose or 0.2% xylose[71]. Where appropriate, media were supplemented with antibiotics at the following concentrations (liquid/solid): kanamycin (5/25 μg/mL); chloramphenicol (2/1 μg/mL); oxytetracycline (2/1 μg/mL); streptomycin (5 μg/mL); spectomycin (25 μg/mL); genta-mycin (1.5 μg/ml). Expression from the vanAB promoter was induced by the addition of a final concentration of 0.5 mM vanillate to cultures.

For carbon starvation experiments, C. crescentus was grown in M2G to OD$_{660}$ of 0.2, washed twice and resuspended in minimal M2 medium without a carbon source. Glucose or xylose was added back to the medium to a final concentration of 0.2% at the desired time point for recovery.

Escherichia coli strains were grown aerobically at 37 °C in LB broth, supplemented with kanamycin (50 μg/mL), chloramphenicol (20 μg/mL), ampicillin (100 μg/ml), gentamycin (20 μg/ml) or tetracyclin

(12 μg/ml) where appropriate. Expression from the P$_{BAD}$ promoter was induced by the addition of arabinose to a final concentration of 0.2%.

## RNA isolation and Northern blot analysis

Bacterial samples were collected with 0.2 volumes of stop-mix (95% ethanol and 5% phenol, vol/vol) and snap-frozen in liquid nitrogen. Total RNA was isolated using the Hot Phenol method as described previously[40] or using the SV Total RNA Isolation System (Promega, #Z3100) following the manufacturer's instructions. Subsequent Northern blot analysis was performed as described before[37,40]. Signals were visualized using a Typhoon Trio™ Scanner (GE Healthcare) and quantified with AIDA Image Analyzer Software (version 5.1., raytest) or Fiji (version 1.53c)[72].

## RT-qPCR

RT-qPCRs were performed in 96-well optical reaction plates (4titude) in biological triplicates on a Bio-Rad CFx96 Touch Real-Time PCR detection system using the commercial Luna® Universal One-Step RT-qPCR kit (NEB, #E3005), according to the manufacturer's recommendations. rsaA mRNA was used as an internal control for RNA normalization.

## Transcriptome analysis by RNA-seq

C. crescentus ΔvanAB or ΔvanAB ΔsisA-D carrying the control vector (pBVMCS-6, ctrl.), CrfA (pP$_{van}$-CrfA) or SisA (pP$_{van}$-SisA), respectively, were grown in triplicates in M2G to OD$_{660}$ of 0.3. Samples were collected 15 minutes after the addition of 0.5 mM vanillate in 0.2 volumes of stop-mix (95% ethanol and 5% phenol, vol/vol) and snap-frozen in liquid nitrogen. RNA was purified with the SV Total RNA Isolation system (Promega, #Z3100), including DNAse-treatment. RNA integrity was confirmed using a Bioanalyzer (Agilent), and rRNA was depleted from the samples using Caulobacter-specific oligos (see Table S1) as described previously[73]. Purified mRNA was subjected to cDNA library preparation with the NEBNext® Ultra™ II Directional RNA Library Prep Kit for Illumina® (NEB, #E7760) according to the manufacturer's instructions, adjusting library size with AMPure XP beads (Beckmann Coulter). Libraries were pooled and sequenced on an Illumina NextSeq1000 system in single-read mode (100 nt read length). Read files in FASTQ format were imported into CLC Genomics Workbench (version 22.0.2., Qiagen), trimmed and mapped to the Caulobacter crescentus NA1000 reference genome (NC_011916.1) using the "RNA-Seq Analysis" tool with standard parameters. Read counts were normalized (CPM) and transformed (log2). Differential expression was tested using the built-in tool corresponding to edgeR in exact mode with tagwise dispersions. Genes with a fold change ≥ 2.0 and an FDR-adjusted p-value ≤ 0.05 were considered as differentially expressed (Supplementary Data 2). The dataset has been deposited at the NCBI GEO repository and is available via the GEO accession GSE275909.

## RIL-Seq

Four replicates of C. crescentus WT and hfq::3xFLAG strains were grown in PYE to an OD$_{660}$ of 1 and processed following the original RIL-seq protocol[38] with modifications. 50 ODs were UV-crosslinked in vivo and after centrifugation (3,500 x g, 20 minutes, 4 °C), the pellet was flash frozen in liquid nitrogen. Cells were lysed by mechanical disruption using the Bead Ruptor (Omni International) with 0.1 mm glass beads, and Hfq-bound RNAs were co-immunoprecipitated using a mono-clonal anti-FLAG antibody (Sigma; #F1804). RNAs were trimmed by RNase A/T1 treatment followed by ligation of proximal RNAs. Upon protein digestion with Proteinase K, RNA was purified, fragmented, and DNase digested. rRNA was depleted as described previously[73]. rRNA-depleted RNA was purified with AMPure XP beads (Beckman-Coulter), and cDNA libraries were prepared for each sample. PCR-amplified cDNA libraries were pooled and sequenced on an Ilumina NextSeq1000 system in a paired-end mode (150 bp read-length). The

de-multiplexed raw reads were analyzed with ChimericFragments[39] using default parameters except for the minimum seed length and the minimum alignment score (min_seed_len=13, min_alignment-_score=17). The resulting table of interactions was filtered according to a minimum read count of 3 and a maximum complementarity FDR of 0.5 (Supplementary Data 1). Correlation between the replicates is documented in Fig. S1d. The genome sequence and annotation of *C. crescentus* used in the analysis were downloaded from NCBI RefSeq (NC_011916.1). The dataset has been deposited at the NCBI GEO repository and is available via the GEO accession GSE283644.

### T7 transcription and 5′ end labelling of RNA

RNA was synthesized in vitro and 5′ end-labelled as described previously[37]. Briefly, PCR amplification was used to generate a DNA template carrying the T7 promoter (KFO-0513/KFO-0514 on gDNA for SisA; KFO-0567/KFO-0568 on gDNA for SisB; KFO-0565/KFO-0566 on gDNA for SisC; KFO-0563/KFO-0564 on gDNA for SisD; KFO-0937/KFO-0938 on gDNA for CrfA; KFO-0937/KFO-0938 on pKF666-1 for CrfA-M2), and RNAs were transcribed using the AmpliScribe T7-Flash transcription kit (Epicentre, #ASF3507). After purification, RNA (20 pmol) was treated with calf intestinal alkaline phosphatase (NEB) for dephosphorylation, and recovered by P:C:I extraction and ethanol precipitation. For 5′ end labelling, dephosphorylated RNA was incubated with $[^{32}P]$-γATP (25 μCi) and polynucleotide kinase (1 unit; NEB) for 1 h at 37 °C. Amersham MicroSpin G-50 columns (Cytiva) were used to remove unincorporated nucleotides. Purification of RNA was then performed by denaturing PAGE on a 6% PAA/7 M urea gel, elution in RNA elution buffer (0.1 M sodium acetate, 0.1% SDS, 10 mM EDTA) at 4 °C overnight, and recovery by P:C:I extraction.

### RNA structure probing

RNA structure probing and mapping of Hfq and sRNA footprints was conducted as described previously[37,74] with some alterations. In short, 0.4 pmol 5′ end-labelled RNA was denatured and mixed with Hfq dilution buffer or *C. crescentus* Hfq (2 pmol or 5 pmol) in a reaction with 1x structure buffer (0.01 M Tris pH 7, 0.1 M KCl, 0.01 M MgCl₂) and 1 μg yeast RNA (Invitrogen, #AM7118). The samples were incubated at 30 °C for 10-15 min, and 4 pmol unlabelled RNA or nuclease-free water was added. Following another incubation at 30 °C for 10–15 min, the samples were treated with RNase T1 (0.1U; Ambion, #AM2283) for 3 min, or lead(II) acetate (final concentration: 5 mM; Sigma-Aldrich, # 316512-5 G) for 1.5 min, respectively. RNase III digests were carried out in the presence of 0.1 mM DTT, added to the 5′ end-labelled RNA before initial denaturation, and treatment with ShortCut RNase III (1.3 U; NEB, #M0245S) for 6 min. All reactions were stopped by adding 2 vol. equiv. precipitation buffer (1 M guanidinium thiocyanate, 0.167% N-laurylsarcosine, 10 mM DTT, 83% 2-propanol) and precipitated at −20 °C overnight. RNA was pelleted, washed with 70% ethanol, and dissolved in GLII loading buffer. RNase T1 and alkaline (OH) sequencing ladders were prepared using 0.8 pmol 5′ end-labelled RNA and reactions stopped by adding 1 vol. equiv. of GLII. Denaturing PAGE on 10% PAA/7 M urea sequencing gels served to separate samples.

### Fluorescence intensity measurements

GFP expression of translational reporter fusions was assayed as described before[37]. In short, *C. crescentus* cells were cultivated overnight in PYE supplemented with the appropriate antibiotics and supplements, *E. coli* cells were diluted and cultivated in LB supplemented with the appropriate antibiotics and supplements for 6 hours. Samples were harvested by centrifugation, washed once in 1x phosphate buffer or 1x PBS, and resuspended in phosphate buffer or PBS. A control sample not expressing GFP was used to determine background fluorescence. Fluorescence intensity in the presence of the sRNA control plasmid was set to 1, and relative expression was calculated from three biological replicates (error bars represent standard deviation).

### Protein sample analysis

Whole cell protein samples were prepared and protein levels analyzed by Western blotting as described previously[40]. Proteins fused to GFP or the 3xFLAG tag were detected using a monoclonal anti-GFP antibody (1:1000; mouse, Roche #11814460001) or anti-FLAG antibody (1:1000; mouse; Sigma #F1804), respectively. GroEL (1:10,000; rabbit; Sigma #G6532) or RNA polymerase (1:5000; mouse; BioLegend #663205) served as loading control. Signals were visualized on a Fusion FX EDGE imager (Vilber) and quantified using Fiji (version 1.53c)[72].

### Bioinformatic tools

Sequence alignments were generated using MultAlin ([75]; version 5.4.1.; http://multalin.toulouse.inra.fr/). RNAfold ([76]; version 2.6.3.; http://rna.tbi.univie.ac.at/cgi-bin/RNAWebSuite/RNAfold.cgi) was employed to predict RNA secondary structures. Predictions for RNA base-pairing interactions were determined with IntaRNA ([77]; version 3.4.1.; http://rna.informatik.uni-freiburg.de/IntaRNA/).

### Statistical analysis and reproducibility

Gene expression from three replicates ($n = 3$), unless stated otherwise, was measured by plate reader or quantified from Western blots or Northern blots. Plots show replicate values, their mean, and standard deviations. Statistical significance was assessed using unpaired two-sided Welch's t-tests (ns for $P > 0.05$, * for $P ≤ 0.05$, ** for $P ≤ 0.01$, *** for $P ≤ 0.001$) with the False Discovery Rate (FDR) approach to correct for multiple comparisons using GraphPad Prism (version 10.0.2., Graphpad software).

Experiments, excluding the RIL-seq and transcriptome analysis, were performed at least twice independently, with up to three biological replicates, to confirm reproducibility of the results.

### Reporting summary

Further information on research design is available in the Nature Portfolio Reporting Summary linked to this article.

## Data availability

The data supporting the findings of this study are available from the corresponding authors upon request. The demultiplexed sequencing data of the RIL-seq experiments have been deposited in the GEO database under accession number GSE283644. The transcriptome analysis by RNA-seq data from this study can be found under the GEO accession code GSE275909. Source data are provided with this paper.

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

## Acknowledgements

We thank members of the Papenfort and Fröhlich labs for project discussions. K.P. acknowledges funding by the German Research Foundation (PA2820/7-1 – Project number 544846468 and EXC 2051 – Project number 390713860) and the European Research Council (ArtRNA, CoG-101088027). Work in the Fröhlich lab is supported by the Deutsche Forschungsgemeinschaft (DFG, German Research Foundation) under Germany´s Excellence Strategy – EXC 2051 – Project number 390713860 and DFG grant FR 3502/2-1 (to K.S.F.). L.N.V. is supported by the Landesgraduiertenstipendium, a scholarship awarded by the Friedrich Schiller University Jena and funded by the State of Thuringia, Germany.

## Author contributions

L.N.V., M.V.G., K.P., and K.S.F. designed the experiments. L.N.V., M.V.G., and K.S.F. performed experiments. L.N.V., M.V.G., M.S. and K.S.F. analysed data. L.N.V. and K.S.F. wrote the manuscript with input from all authors.

## Funding

## Competing interests

The authors declare no competing interests.
