## [Transparent Peer review file · Nature Communications]

An RNA sponge directs the transition from feast to famine in *Caulobacter crescentus*

Corresponding Author: Dr Kathrin Fröhlich

Version 0:

Reviewer comments:

Reviewer #1

(Remarks to the Author)

This manuscript presents a well-executed study on RNA-based regulation in *Caulobacter crescentus*, revealing how the carbon starvation-induced small RNA CrfA functions as a sponge to inactivate a family of sibling sRNAs (SisA-D), thereby reprogramming gene expression in response to changing nutrient availability. While previous work had identified CrfA as an important sRNA regulator, it was assumed to directly pair with and activate a set of targets involved in carbon transport. Using RIL-seq, in which pairs of sRNAs and pairing target mRNAs are ligated to identify chimeric RNAs, reflecting global RNA-RNA interactions, the authors show that CrfA, instead of making chimeras with the targets, was found to base-pair with a set of similar sRNAs, the most abundant of which is SisA. SisA (and the other Sis RNAs) pair with and negatively regulate the apparent CrfA targets; CrfA activates by relieving SisA repression. The data convincingly demonstrate that this regulatory mechanism is essential for *Caulobacter*'s adaptation to fluctuating carbon sources, with loss of CrfA leading to severe growth defects under stress. The manuscript is very well written, with a clear and engaging narrative. The background is thorough yet accessible, and the conclusions are strongly supported by the results. The issues raised below are to provide a bit more clarification and suggest a couple of additional tests.

Specific issues:

1. Fig. 1a: Is it useful to show this distribution for all RNAs, not just significant ones? This might be moved to supplemental material. While it is clear that chimeras are enriched significantly by the 3xFlag-Hfq vs. WT, (100x up), the singles are much less increased. Is this as high on WT as one would expect?
2. Lines 113-121: Of the 1535 statistically significant chimeras, only about half (607) were retained by the Chimeric Fragments pipeline. What gets left out by this? It might be useful to have the full list for reference (all statistically significant) for readers to have those to consider, understand what is removed by this program, which seems to require pairing. I don't think this program has been used in other RIL-Seq datasets. Are the orientations similar as the full set in terms of RNA1 vs. RNA2? Anything interesting in terms of what was left out? Is having 146/607 with at least one sRNA what one would expect?
3. Table S1: Please label CrfA as well as the other Sis genes in this table. For other entries that are listed as ncRNAs in position 1, it was sometimes difficult to understand what they are. Where would that be listed? For instance, 3_CCNA_02328; is this 3' UTR of gcrA? For others (for instance R0210), it is listed in NCBI as antisense and is shown with a pair that is antisense to that. Should that be indicated someplace as a category? In any case, an appropriate place to find all of these entries should be indicated in the table legend.
4. Given the large number of SisA, CrfA chimeras, what would sRNA 1 or 2 look like aside from these? All of CrfA are in position 1; are almost all position ones that are ncRNAs pairing with another ncRNA (other decoys)?
5. Fig. S3e: The alignment of the upstream region of these genes appears, as noted to have nothing in common. Is there no core consensus sequence for *Caulobacter* anywhere upstream of these sRNA starts (with processing afterwards)? While they could have different activators or repressors, it seems surprising not to have any promoter regions.
6. Fig. S4 expression/levels: The results reinforce the idea that these different sRNAs have somewhat different patterns, presumably in part due to synthesis levels and in part degradation. In particular, does the pattern look very different in the absence of CrfA?
7. Fig. 2b-d: Please check legends for c, d (appear to be mixed up),
8. Fig. 3: A bit more text to discuss these results would be useful. Does loss of Hfq have less effect on sRNA levels in *Caulobacter* than seen in some other organisms? Fig. 3d measures CrfA levels, which do still seem to decrease without Hfq when SisA is expressed (last bar significantly lower than control). SisA is more affected by CrfA, but in this case, only in

hfq+. Is it surprising these are not reciprocal?

9. Fig. 4: This is key experiment in demonstrating the epistasis, showing that SisA is needed for CrfA-dependent regulation, but not vice versa. Considering spelling this out a bit more.

10. Fig. 4b, c: Is it necessary to use Hfq from *Caulobacter*? If so, that might be worth mentioning. *E. coli* Hfq is still present, correct? Hfq from *Caulobacter* overexpressed (please clarify in legend)? As above, this shows that SisA is sufficient for regulation. It would be good to also test CrfA. Presumably it does nothing? This would further confirm the epistasis in 4a.

11. Fig. 5: Overexpression of CrfA seems to have some effect (although much weaker), on the same genes in the absence of any sis genes. Is there an explanation for this? Another Sis-like mRNA, or possibly a second role of CrfA (by pairing or maybe encoding a short, active peptide)? Any reason to believe CrfA might bind/interfere with other roles of Hfq? Do any of the mRNAs that still show some CrfA effect have non-Sis chimeras? While this may all be beyond the scope of this work, it seems possible that such a biologically important RNA might be doing more things than the Sis decoy activity.

12. Fig. 7: Is it known what would grow up (what suppressors, for instance) in *del crfA* in xylose? Are the transporters for xylose known?

13. Why was OD660 used some places, OD600 others?

14. Some suggestions for the discussion:

a. A bit more discussion of how CrfA likely interacts with Hfq and serves as a decoy or sponge would be useful. CrfA doesn't have a polyU tail, so may be more like a target and was entirely at position 1. Possibly this could be explicitly mentioned.

b. I would be inclined to more clearly discuss how CrfA was originally mis-interpreted (I think mostly that the pairing wasn't so clear, and the appropriate compensating mutations never tested?).

c. The authors do not currently refer to Figure 8 (the model figure) in the main text.

d. Adding subheadings in the Discussion section could help structure the interpretation of findings and improve clarity for readers.

Reviewer #2

(Remarks to the Author)

Reviewer #3

(Remarks to the Author)

In this study, Vogt and Velasco-Gomariz et al. examined the RNA-RNA interactome associated with the *Caulobacter crescentus* Hfq RNA chaperone protein using the RIL-seq approach. The authors detected many interactions between small RNAs (sRNAs) and other RNAs, of which a large percentage include the SisA sRNA. Most importantly, the Vogt and Velasco-Gomariz et al. found that the CrfA sRNA previously found to affect metabolism in response to carbon-starvation does not directly base pair with the reported targets but instead serves as a sponge RNA to downregulated constitutively-expressed SisA and the other aR8 RNA family RNAs SisB, SisC and SisD. Consistent with the CrfA role in regulating carbon metabolism through SisA, strains lacking the sponge RNA had growth defects in environments with fluctuating carbon levels.

The CrfA, SisA, SisB, SisC and SisD are interesting sRNAs, and the data are convincing. I have relatively minor comments:

1. The first part of the manuscript jumps around unnecessarily. Some interrelated suggestions:

--I suggest that the "story line" of the paper be about how CrfA was mischaracterized. The RIL-seq data showed that CrfA is a sponge!

--Considering the first point, there should be more information about CrfA as well as about sponge RNAs in the introduction. The finding that CrfA is a sponge RNA should be stated explicitly in the introduction. Similarly, I think the first few sentences of the results describing published findings make more sense in the introduction. Instead, the long paragraph about the transcriptional regulators, which do not feature in the results should be condensed.

2. The Sis family of RNAs is intriguing. Given this the authors should:

--Page 5, line 141: Define and describe the aR8 RNA family at the first mention.

--Include more of the findings about SisB, C and D (such as northern analysis) in the main figures.

3. The authors should be explicit about experimental details such as whether the CCNA_03574::gfp is on a plasmid or the chromosome (Page 7, line 195).

4. Some wording could be "toned down" without detracting from the paper.

--Page 2, line 27: sRNA-sRNA pairs or sponge RNAs are no longer so "unconventional".

--Page 13, line 403: Given the findings described, the Qrr family no longer "has a peculiarity".

5. Additional editorial suggestions:

--Page 3, line 50: "destined"? Perhaps replace with "utilized".

--Page 4, line 103: Replace "co-immunoprecipitated from Hfq" with "co-immunoprecipitated with Hfq".

--Page 5, lines 119-121: The numbers here are somewhat confusing. Are the authors referring to one third of 607 or of 146?

- Page 5, lines 131-133: The sentence beginning "In addition to these mRNA targets,..." is a little awkward.
- Page 6, line 145: Does CCNA_00780 encode a glutamate amidohydrolase?
- Page 10, lines 319-321: Suggest "...the functional context in which distinct sRNAs are embedded". Here, we report RIL-seq mapping of the Hfq-dependent RNA-RNA network in *C. crescentus*."
- Page 11, line 332: Replace "alternative" with "multiple".

Reviewer #4

(Remarks to the Author)

In this manuscript, Vogt and co-workers employ the RIL-seq approach to globally identify RNA-RNA interactions associated with the Hfq protein in *Caulobacter crescentus*. From the hundreds of identified interactions, the authors focus on the association of the partially characterized sRNA CrfA and the four sequence-related sibling sRNAs SisA-D. The authors reveal that CrfA acts as a sponge of the Sis sRNAs, thereby relieving Sis-dependent regulation of a large number of mRNA targets connected to nutrient metabolism. The CrfA-Sis interaction is partially supported by mutational analysis in vivo and structure probing in vitro. SisA-dependent regulation of mRNA targets is convincingly determined through compensatory mutation experiments. Importantly, the authors show that loss of CrfA leads to a severe growth phenotype during recovery from carbon starvation.

This is a very well-written manuscript presenting data of high quality. It is an important contribution to the field, especially since it presents an example of sRNA-based regulation with strong impact on bacterial proliferation. Still, prior to publication, there are some issues that need to be addressed, as specified below.

Specific comments:

1. The RIL-seq data is based on four biological replicates. Please inform the reader about the reproducibility between the replicates, including overlap of chimeras and statistical correlation.
2. The authors convincingly show that CrfA-dependent regulation of mRNA translation, which previously has been suggested to be direct, in fact is indirect and through the Sis sRNAs (e.g. Fig. 4a). In Figure 3, the authors show that overexpression of CrfA leads to reduced SisA levels, and that overexpression of SisA leads to reduced CrfA levels. A point mutation in the predicted interaction site in either CrfA or SisA leads to reduced effect on the other sRNA. However, although the M1 mutants of either sRNA were designed to restore complementarity when combined, the authors did not show whether restoring complementarity also restores regulation. Thus, whether the two sRNAs indeed interact, and whether this interaction is required for regulation, still remains somewhat an open question.
3. A direct interaction between SisA and CrfA is further investigated in vitro using structure probing. The experiment shown in Fig. 2e show clear protection from lead cleavage of CrfA in the presence of Sis sRNAs. However, the reciprocal experiment where SisA cleavage is monitored in the presence of CrfA is less convincing (Fig. S6). The text referring to Fig. S6 reads "we detected base-pairing of CrfA to SisA....covering the conserved 10-mer sequence core", but any information about how the result in Fig. S6 supports this claim is lacking. In fact, there is no change in lead reactivity in the predicted CrfA binding site of SisA when comparing the presence or absence of SisA. Since the predicted binding site is in a region that according to Fig. 2b is single-stranded (nt 37-46 of SisA), CrfA binding in this region should result in clear protection from cleavage. Instead, there is a clear protection from lead cleavage around position 55 when Hfq is added, but this is not mentioned in the text. This not only casts doubt over the CrfA-SisA interaction, but also whether the structure representation shown in Fig. 2b is correct. Together, the in vivo and in vitro analysis of CrfA-SisA interaction needs to be more convincingly demonstrated. For instance, did the authors perform EMSAs with SisA-CrfA using the M1 mutants? Does co-expression of M1 versions of CrfA and SisA result in regulation of target mRNA translation, while WT/mutant combinations do not?

Minor comments:

1. Figure 1: The numbers to the right of staples in panels a and b are neither explained in the figure nor in the legend. I reckon this is the number of chimeras, but it should be clearly stated. In a, the writing "2.5 E05" may be unclear for some readers. Please use a more mathematically correct writing instead. In d, the font size for chromosomal positions needs to be increased.
2. Sentence starting on line 131 "In addition to..." should include a reference to a figure or table.
3. Fig. 2b: Please add numbers in the structure representation for SisA. This is required for the reader to be able to evaluate the results of Fig. S6. Alternatively, add the structure with numbering next to the gel in Fig. S6.
4. Line 164 and Fig. 2c: The text states that the M2 mutation replaces residues G49-G54 with "a non-complementary tetraloop sequence". By contrast, the annotation in Fig. 2c gives the impression that those nucleotides simply were deleted (delta sign). Please clarify.
5. Lines 179-180 and Fig. 3d: The text claims that deletion of Hfq abrogated the inhibitory effect of CrfA and SisA on each other. However, Fig. 3d shows practically no difference in CrfA levels upon SisA overexpression with respect to the

presence or absence of Hfq. In both cases, SisA overexpression leads to a significant reduction of CrfA. Please change the text to properly describe the presented data.

6. Figure 5: I would highly recommend the authors to change the heat-map representation to a bar graph. This would make it easier to evaluate the differences between CrfA overexpression in "WT" and sisAD deletion strains.

7. Line 307: the wording "the change in routine resulted in..." is unclear. Please rephrase to precisely describe what is meant.

Version 1:

Reviewer comments:

Reviewer #1

(Remarks to the Author)

In this revision of a study of the small RNA network in *Caulobacter crescentus*, the authors have done a very nice job of answering questions raised by us and other reviewers, and have clarified what was done at various stages. The manuscript will be important both to those studying *Caulobacter* and how it responds to changing carbon sources and those interested in sRNA-based regulation; it also serves as a cautionary note on interpreting control by sRNAs.

Some minor issues that can easily be corrected:

1. Line 111: The total chimeric number 493,886 does not match what is shown in Fig. 1a (484,969). Clarification is needed.
2. Legend of Fig. S7a: CfrA-M1 (Pconst::crfA-M1) was not tested and should be removed from the legend.
3. Figs. S7c/d were not mentioned in the main text. Either remove the figures or add them somewhere in the text.

Reviewer #2

(Remarks to the Author)

Reviewer #4

(Remarks to the Author)

The authors have convincingly addressed all comments.

We would like to thank the reviewers for the careful evaluation of our manuscript. We have addressed all comments and highlighted changes in the manuscript.

RESPONSE TO REVIEWER COMMENTS

Reviewer #1 (Remarks to the Author):

This manuscript presents a well-executed study on RNA-based regulation in *Caulobacter crescentus*, revealing how the carbon starvation-induced small RNA CrfA functions as a sponge to inactivate a family of sibling sRNAs (SisA-D), thereby reprogramming gene expression in response to changing nutrient availability. While previous work had identified CrfA as an important sRNA regulator, it was assumed to directly pair with and activate a set of targets involved in carbon transport. Using RIL-seq, in which pairs of sRNAs and pairing target mRNAs are ligated to identify chimeric RNAs, reflecting global RNA-RNA interactions, the authors show that CrfA, instead of making chimeras with the targets, was found to base-pair with a set of similar sRNAs, the most abundant of which is SisA. SisA (and the other Sis RNAs) pair with and negatively regulate the apparent CrfA targets; CrfA activates by relieving SisA repression. The data convincingly demonstrate that this regulatory mechanism is essential for *Caulobacter*'s adaptation to fluctuating carbon sources, with loss of CrfA leading to severe growth defects under stress. The manuscript is very well written, with a clear and engaging narrative. The background is thorough yet accessible, and the conclusions are strongly supported by the results. The issues raised below are to provide a bit more clarification and suggest a couple of additional tests.

Specific issues:

1. Fig. 1a: Is it useful to show this distribution for all RNAs, not just significant ones? This might be moved to supplemental material. While it is clear that chimeras are enriched significantly by the 3xFlag-Hfq vs. WT, (100x up), the singles are much less increased. Is this as high on WT as one would expect?

We have decided to show the distribution of all RNA species to outline our data analysis pipeline. As observed by the reviewer, chimeric reads (~70x) are twice more strongly enriched compared to single reads (~35x). We do not know whether that reflects any differences e.g. in stability of RNA species associated with the specific purification of Hfq-bound RNAs vs. the unspecific recovery on untagged Hfq protein in the wild-type sample but we have observed a similar discrepancy for a RIL-seq experiment in *Klebsiella pneumoniae* performed in our lab (Ruhland et al. 2024; PMID 38377209) but also in independent RIL-seq studies performed by other groups in *E. coli* (Melamed et al. 2016; PMID 27588604) or *S. Typhimurium* (Matera et al. 2022; PMID 35063132), suggesting that this phenomenon is a typical feature of the methodology and not specific to our study.

2. Lines 113-121: Of the 1535 statistically significant chimeras, only about half (607) were retained by the Chimeric Fragments pipeline. What gets left out by this? It might be useful to have the full list for reference (all statistically significant) for readers to have those to consider, understand what is removed by this program, which seems to require pairing. I don't think this program has been used in other RIL-Seq datasets. Are the orientations similar as the full set in terms of RNA1 vs. RNA2? Anything interesting in terms of what was left out? Is having 146/607 with at least one sRNA what one would expect?

The ChimericFragments pipeline (Siemers et al.; PMID 38633425) assigns an additional significance value based on the complementarity of the two RNAs in a chimeric read around the ligation point considered to be in proximity to the likely interaction site. In addition to benchmarking in the original publication, ChimericFragments was also applied to RIL-seq datasets for *Vibrio cholerae* ProQ (Ghandour et al. 2025; PMID 39727155), and to Hfq RIL-seq in *K. pneumoniae* (Ruhland et al. 2024; PMID 38377209). By applying this workflow to our dataset, we filtered for high confidence chimeras with good base-pairing predictions. This excluded mainly mRNA-mRNA and mRNA-IGR chimeras, as well as some sRNA-mRNA chimeras, many of which involved sRNAs antisense to transposases. The 928 chimeras that were filtered out also included 43 additional SisA chimeras and 7 CrfA-mRNA chimeras, 6 of which had the sRNA at position 1 as in its chimeras with SisA-D. The full list of statistically significant chimeras was added as an additional sheet to Table S1.

For high confidence chimeras, the distribution of RNA classes for position 1 and position 2 barely changes compared to the distribution of all significant chimeras (s. Fig. 1), e.g. 27% of sRNAs involved in high confidence chimeras are at position 1, which is a small decrease compared to 31% of sRNAs in all significant chimeras at position 1.

Figure 1: Relative distribution of RNA classes in significant chimeras (left) and high-confidence chimeras (right).

Around 25% of high confidence chimeras involve at least one sRNA. This fraction is smaller compared to what has been recovered in *E. coli* (40%, Melamed et al. 2016; PMID 27588604) or *K. pneumoniae* (47%, Ruhland et al. 2024; PMID 38377209), but slightly higher than in *Clostridioides* (20%, Fuchs et al. 2023 EMBO J; PMID 37140366). It is therefore likely that these differences are a species-specific characteristic.

3. Table S1: Please label CrfA as well as the other Sis genes in this table. For other entries that are listed as ncRNAs in position 1, it was sometimes difficult to understand what they are. Where would that be listed? For instance, 3_CCNA_02328; is this 3' UTR of gcrA? For others (for instance R0210), it is listed in NCBI as antisense and is shown with a pair that is antisense to that. Should that be indicated someplace as a category? In any case, an appropriate place to find all of these entries should be indicated in the table legend.

Thank you for this feedback on the labels and annotations in Table S1. The nomenclature was based on the current NCBI annotation of the *C. crescentus* NA1000 genome. We have renamed the

original CCNA_R0xxx annotation as SisA-D and CrfA. We clarified the 3_CCNA_xxxx nomenclature as 3'-derived sRNAs in the table legend and added a column indicating 3'-derived and antisense sRNAs.

4. Given the large number of SisA, CrfA chimeras, what would sRNA 1 or 2 look like aside from these? All of CrfA are in position 1; are almost all position ones that are ncRNAs pairing with another ncRNA (other decoys?)?

Even though the CrfA-SisA interaction is the most abundant regarding the number of reads, it is represented only by one unique chimera. Since our distribution analyses for Fig. 1b/c and S1c are based on the count of unique chimera and not the number of reads of those chimeras, excluding the CrfA-SisA interaction does not change the relative distribution of sRNAs at position 1 or position 2.

When examining our high confidence chimeras for possible other decoys, we find 41 chimeras with 27 different sRNAs in position 1. Most of the chimeras recovered with sRNAs in position 1 have an mRNA (22 of 41) or IGR (8 of 41) transcript mapped at position 2. Of the 27 sRNAs, only five exclusively have chimeric interactions with other sRNAs, adding up to 11 sRNA-sRNA chimeras for sRNAs in position 1 (27%). This includes CrfA (chimeras with SisA-D), the pair R0078-R0210 (in both orientations) with R0210 encoded as an antisense RNA to R0078, R0018 in a chimeric interaction with the partially overlapping antisense encoded R0019, and R0188 in a chimera with SisA. Whether these sRNA-sRNA interactions suggest decoy activities beyond typical antisense sRNAs has to be experimentally addressed.

5. Fig. S3e: The alignment of the upstream region of these genes appears, as noted to have nothing in common. Is there no core consensus sequence for *Caulobacter* anywhere upstream of these sRNA starts (with processing afterwards)? While they could have different activators or repressors, it seems surprising not to have any promoter regions.

As pointed out by the reviewer, a common binding site for the polymerase should be detectable at a distinct site upstream the transcription start site.

The motif for RpoD-binding in *Caulobacter* is relatively weak (see Figure; PMID 25569173); however, we do find the TTG motif at the expected distance (centred at -35) from the TSS for all sibling sRNAs. We have updated our alignment to highlight this information (see updated Figure S3).

6. Fig. S4 expression/levels: The results reinforce the idea that these different sRNAs have somewhat different patterns, presumably in part due to synthesis levels and in part degradation. In particular, does the pattern look very different in the absence of CrfA ?

All four sibling sRNAs have very distinct expression patterns, and as stated in the manuscript we think that different transcription factors are involved in their synthesis. CrfA likely plays a role in the decay of the Sis family RNAs and its deletion does mildly increase the expression maxima under standard conditions compared to the wild-type (see new Fig. S4b).

7. Fig. 2b-d: Please check legends for c, d (appear to be mixed up)

Thank you for pointing out this mistake, we have adjusted the legend for Fig. 2c-d.

8. Fig. 3: A bit more text to discuss these results would be useful. Does loss of Hfq have less effect on sRNA levels in *Caulobacter* than seen in some other organisms? Fig. 3d measures CrfA levels, which do still seem to decrease without Hfq when SisA is expressed (last bar significantly lower than control). SisA is more affected by CrfA, but in this case, only in hfq+. Is it surprising these are not reciprocal?

Compared to other Gram-negative species like *E. coli*, we observe on average less dramatic destabilization of sRNAs in the absence of Hfq in *Caulobacter* for the examples we have addressed (e.g. Vogt et al. 2024; PMID 38511926).

We have carefully rephrased this section (see also comment 5 of reviewer 4) as there is a reduced but detectable regulation in the absence of Hfq. We have newly added an EMSA which also indicates complex formation between CrfA and SisA in the absence of Hfq, although less efficient when compared to the reaction containing the protein (new Figure S6d).

We currently have not addressed the fate of CrfA and SisA when interacting with each other, and further studies will be needed to understand the competition between CrfA and the mRNA targets of SisA. We now added a comment regarding these points to the discussion.

9. Fig. 4: This is key experiment in demonstrating the epistasis, showing that SisA is needed for CrfA-dependent regulation, but not vice versa. Considering spelling this out a bit more.

We have added additional information on how CrfA was initially misinterpreted to emphasize the respective role of SisA and the sponge.

10. Fig. 4b, c: Is it necessary to use Hfq from *Caulobacter*? If so, that might be worth mentioning. *E. coli* Hfq is still present, correct? Hfq from *Caulobacter* overexpressed (please clarify in legend)? As above, this shows that SisA is sufficient for regulation. It would be good to also test CrfA. Presumably it does nothing? This would further confirm the epistasis in 4a.

We have previously shown that *Caulobacter* Hfq binds RNA with lower affinity and different specificity compared to *E. coli* Hfq (PMID 31076551). For our target validation experiments in *E. coli* we therefore used a strain that expresses *Caulobacter* Hfq as the sole copy under control of the endogenous *E. coli* *hfq* promoter (PMID 31076551). In addition to the methods section we have now added the description of the strain in the figure legend of Fig. 4c.

We agree with the reviewer that it is a good idea to test the effect of CrfA on the fusion. Interestingly, the CrfA transcript fails to terminate efficiently in *E. coli*. We detect a run-through product terminating only at the internal terminator included on the plasmid (not shown) which is significantly longer than the sRNA to be tested. To overcome this issue, we have constructed a

version in which we attached a self-cleaving ribozyme to the 3' end of *crfA* on the *E. coli* expression plasmid. With this construct, we minimize run-through and produce full-length CrfA. We show that, as expected, CrfA in *E. coli* does not affect target reporter expression which confirms that SisA alone directly regulates expression of *CCNA_03574* (data shown in new Figure 4d; sRNA expression shown in new Figure S8).

11. Fig. 5: Overexpression of CrfA seems to have some effect (although much weaker), on the same genes in the absence of any sis genes. Is there an explanation for this? Another Sis-like mRNA, or possibly a second role of CrfA (by pairing or maybe encoding a short, active peptide)? Any reason to believe CrfA might bind/interfere with other roles of Hfq? Do any of the mRNAs that still show some CrfA effect have non-Sis chimeras? While this may all be beyond the scope of this work, it seems possible that such a biologically important RNA might be doing more things than the Sis decoy activity.

We agree with the reviewer that CrfA and the Sis sRNAs have ample potential for future studies. We currently have no evidence to support the hypothesis that CrfA may serve additional roles besides the regulation of the Sis sRNAs. A preliminary bioinformatics analysis of *crfA* conservation suggests that the sponge RNA is only present in species with at least one Sis RNA (unpublished). However, it is indeed intriguing that some transcripts were mildly deregulated in response to CrfA pulse-overexpression also in the absence of SisA-D. One possible explanation is an indirect effect on Hfq availability as the strong pulse induction of CrfA in this experiment could at least partially titrate the RNA chaperone, resulting in a short-term deregulation of the mRNAs. We would expect that the cell is able to balance this effect with time as we do not detect it in continuous overexpression. Indeed, the top four upregulated mRNA transcripts in response to CrfA pulse overexpression (*CCNA_03181*, *CCNA_03263*, *CCNA_03574*, *meaA* and *malA*), were significantly downregulated by SisA while CrfA had no effect on GFP levels (*03181* (not shown), *03263* (s. Fig. S9A), *03574* (s. Fig. 4)) or only caused mild repression (*meaA*, 1.6-fold (not shown)). Regulation of *malA* was not addressed.

In regard of the other questions in this comment:

- A ribosome-profiling experiment (PMID 25078267) did not reveal translation of the *crfA* transcript, we therefore currently exclude the possibility that CrfA is a dual function RNA. We cannot exclude the interaction of CrfA with another, currently unknown RBP but Hfq.
- Further high-confidence chimeras were found only between *meaA* and *CCNA_01805*, *malA* and *CCNA_03360*, and *malA* and *CCNA_02458*, respectively. We currently do not know whether these mRNA-mRNA pairs have any regulatory role.

12. Fig. 7: Is it known what would grow up (what suppressors, for instance) in *del crfA* in xylose? Are the transporters for xylose known?

We currently have not selected for mutants that enhance growth in the presence of xylose in a *crfA* deletion background. One could speculate that the inactivation of the xylose transport would rescue the phenotype but the uptake of this sugar is most likely not linked to a single gene/system. When Stephens et al. (2006) identified genes required for xylose utilization in *C. crescentus*, the transposon mutagenesis did not reveal a xylose transporter, indicating a potential redundancy in sugar uptake (PMID 17172333). In a follow-up, the authors showed that expression of *XylE*, a

predicted protein similar to the *E. coli* xylose:H⁺ symporter, was under control of the XylR repressor that regulates expression of the enzymes required for xylose metabolism *xylXABCD* (2007, PMID 17933895). Furthermore, *C. crescentus* encodes more than 60 TonB-dependent receptors (TBDRs) for which the individual substrates are currently mostly unknown. The uptake of xylose and xylo-oligosaccharides through TBDRs has been demonstrated for other bacteria, and it is therefore well possible that one or more transporters of this type also contribute to xylose uptake in *C. crescentus*. For example, the two TBDRs CCNA_01051 and CCNA_02923 are induced in the presence of xylose, however their function is yet to be determined (PMID 14973021).

13. Why was OD660 used some places, OD600 others?

The optical density of *Caulobacter* is typically measured at 660 nm, while *E. coli* optical density is measured at 600 nm. Thus, we use OD₆₀₀ in the legend of Figure S7b describing an experiment performed in *E. coli*.

14. Some suggestions for the discussion:

a. A bit more discussion of how CrfA likely interacts with Hfq and serves as a decoy or sponge would be useful. CrfA doesn't have a polyU tail, so may be more like a target and was entirely at position 1. Possibly this could be explicitly mentioned.

We have added a remark in the discussion to comment on the position of CrfA on Hfq as well as the potential competition with mRNA targets of the sibling sRNAs.

b. I would be inclined to more clearly discuss how CrfA was originally mis-interpreted (I think mostly that the pairing wasn't so clear, and the appropriate compensating mutations never tested?).

As assumed by the reviewer, the mechanism of positive regulation of CCNA_3574 mRNA (or any of the other transcripts identified to be upregulated) by CrfA had not been addressed in the original publication on this sRNA, and the potential base-pairing was not confirmed by a compensatory exchange. The originally used CrfA mutant was predicted to interrupt mRNA recognition but instead is positioned within the sibling sRNA binding site, and therefore failed to confer upregulation. We have added information into the discussion.

c. The authors do not currently refer to Figure 8 (the model figure) in the main text.

Thank you – we have fixed that issue and now refer to the figure.

d. Adding subheadings in the Discussion section could help structure the interpretation of findings and improve clarity for readers.

We did not include subheadings in the discussion as per Nature Communications guidelines.

Reviewer #2 (Remarks to the Author):

Reviewer #3 (Remarks to the Author):

In this study, Vogt and Velasco-Gomariz et al. examined the RNA-RNA interactome associated with the *Caulobacter crescentus* Hfq RNA chaperone protein using the RIL-seq approach. The authors detected many interactions between small RNAs (sRNAs) and other RNAs, of which a large percentage include the SisA sRNA. Most importantly, the Vogt and Velasco-Gomariz et al. found that the CrIA sRNA previously found to affect metabolism in response to carbon-starvation does not directly base pair with the reported targets but instead serves as a sponge RNA to downregulated constitutively-expressed SisA and the other aR8 RNA family RNAs SisB, SisC and SisD. Consistent with the CrIA role in regulating carbon metabolism through SisA, strains lacking the sponge RNA had growth defects in environments with fluctuating carbon levels.

The CrIA, SisA, SisB, SisC and SisD are interesting sRNAs, and the data are convincing. I have relatively minor comments:

1. The first part of the manuscript jumps around unnecessarily. Some interrelated suggestions:
--I suggest that the "story line" of the paper be about how CrfA was mischaracterized. The RIL-seq data showed that CrfA is a sponge!
--Considering the first point, there should be more information about CrfA as well as about sponge RNAs in the introduction. The finding that CrfA is a sponge RNA should be stated explicitly in the introduction. Similarly, I think the first few sentences of the results describing published findings make more sense in the introduction. Instead, the long paragraph about the transcriptional regulators, which do not feature in the results should be condensed.

As suggested by the reviewer we have added a section on RNA sponges to the introduction and cut the paragraph in the beginning of the results. We have also shortened the paragraph on *Caulobacter* sRNAs.

2. The Sis family of RNAs is intriguing. Given this the authors should:
--Page 5, line 141: Define and describe the aR8 RNA family at the first mention.

We are not entirely sure how to interpret this comment. We added a remark to the introduction to point to the affiliation of SisA-D with the aR8 family. The description and definition of SisA-D is at the indicated position.

--Include more of the findings about SisB, C and D (such as northern analysis) in the main figures.

We have added a Northern blot analysis on the differential expression of the Sis family to main Figure 2.

3. The authors should be explicit about experimental details such as whether the CCNA_03574::gfp is on a plasmid or the chromosome (Page 7, line 195).

We apologize if our description was misleading. Our reporter system in *Caulobacter* consists of a plasmid-encoded sRNA and a target mRNA::gfp fusion. The latter construct is always integrated

into the same, neutral locus on the chromosome (*rsaA* gene; not in the target gene). We have added a remark to the figure legend and the method section.

4. Some wording could be “toned down” without detracting from the paper.

--Page 2, line 27: sRNA-sRNA pairs or sponge RNAs are no longer so “unconventional”.

Done.

--Page 13, line 403: Given the findings described, the Qrr family no longer “has a peculiarity”.

Done.

5. Additional editorial suggestions:

--Page 3, line 50: “destined”? Perhaps replace with “utilized”.

Done.

--Page 4, line 103: Replace “co-immunoprecipitated from Hfq” with “co-immunoprecipitated with Hfq”.

Done.

--Page 5, lines 119-121: The numbers here are somewhat confusing. Are the authors referring to one third of 607 or of 146?

Done.

--Page 5, lines 131-133: The sentence beginning “In addition to these mRNA targets,...” is a little awkward.

Done.

--Page 6, line 145: Does CCNA_00780 encode a glutamate amidohydrolase?

Yes. We now emphasize that.

--Page 10, lines 319-321: Suggest “..the functional context in which distinct sRNAs are embedded⁴⁴. Here, we report RIL-seq mapping of the Hfq-dependent RNA-RNA network in *C. crescentus*.”

Done.

--Page 11, line 332: Replace “alternative” with “multiple”.

Done.

Reviewer #4 (Remarks to the Author):

In this manuscript, Vogt and co-workers employ the RIL-seq approach to globally identify RNA-RNA interactions associated with the Hfq protein in *Caulobacter crescentus*. From the hundreds of identified interactions, the authors focus on the association of the partially characterized sRNA CrfA and the four sequence-related sibling sRNAs SisA-D. The authors reveal that CrfA acts as a sponge of the Sis sRNAs, thereby relieving Sis-dependent regulation of a large number of mRNA targets connected to nutrient metabolism. The CrfA-Sis interaction is partially supported by mutational analysis *in vivo* and structure probing *in vitro*. SisA-dependent regulation of mRNA targets is convincingly determined through compensatory mutation experiments. Importantly, the authors show that loss of CrfA leads to a severe growth phenotype during recovery from carbon starvation.

This is a very well-written manuscript presenting data of high quality. It is an important contribution to the field, especially since it presents an example of sRNA-based regulation with strong impact on bacterial proliferation. Still, prior to publication, there are some issues that need to be addressed, as specified below.

Specific comments:

1. The RIL-seq data is based on four biological replicates. Please inform the reader about the reproducibility between the replicates, including overlap of chimeras and statistical correlation.

We have added information about the overlap between the individual replicates (Fig. S1d). The most abundant chimeras (like SisA-CrfA) are recovered by all four experiments. The first two datasets show less overlap, and one reason might be the lower sequencing depth that was chosen when the method was to be established in the lab. We have nevertheless decided to include results of all four experiments to maximize the number of interactions as a resource.

2. The authors convincingly show that CrfA-dependent regulation of mRNA translation, which previously has been suggested to be direct, in fact is indirect and through the Sis sRNAs (e.g. Fig. 4a). In Figure 3, the authors show that overexpression of CrfA leads to reduced SisA levels, and that overexpression of SisA leads to reduced CrfA levels. A point mutation in the predicted interaction site in either CrfA or SisA leads to reduced effect on the other sRNA. However, although the M1 mutants of either sRNA were designed to restore complementarity when combined, the authors did not show whether restoring complementarity also restores regulation. Thus, whether the two sRNAs indeed interact, and whether this interaction is required for regulation, still remains somewhat an open question.

As suggested by the reviewer we have now added an experiment to show that compensatory mutations in SisA and CrfA confirm regulation via a direct interaction of the two sRNAs (Fig. 3c/d).

3. A direct interaction between SisA and CrfA is further investigated *in vitro* using structure probing. The experiment shown in Fig. 2e show clear protection from lead cleavage of CrfA in the presence of Sis sRNAs. However, the reciprocal experiment where SisA cleavage is monitored in the presence of CrfA is less convincing (Fig. S6). The text referring to Fig. S6 reads “we detected base-pairing of CrfA to SisA....covering the conserved 10-mer sequence core”, but any information about how the result

in Fig. S6 supports this claim is lacking. In fact, there is no change in lead reactivity in the predicted CrfA binding site of SisA when comparing the presence or absence of SisA. Since the predicted binding site is in a region that according to Fig. 2b is single-stranded (nt 37-46 of SisA), CrfA binding in this region should result in clear protection from cleavage. Instead, there is a clear protection from lead cleavage around position 55 when Hfq is added, but this is not mentioned in the text. This not only casts doubt over the CrfA-SisA interaction, but also whether the structure representation shown in Fig. 2b is correct. Together, the *in vivo* and *in vitro* analysis of CrfA-SisA interaction needs to be more convincingly demonstrated. For instance, did the authors perform EMSAs with SisA-CrfA using the M1 mutants? Does co-expression of M1 versions of CrfA and SisA result in regulation of target mRNA translation, while WT/mutant combinations do not?

As this comment consists of multiple questions/remarks we have split our response.

- **In vitro experiments presented in 6a:**
We have now indicated the potential Hfq binding site in SisA in Fig. S6a. While lead cleavage of SisA in the presence of CrfA is mildly altered at the expected binding site we do see a clear change in the RNase III cleavage pattern with increased reactivity around C42 within the 10-mer sequence core, indicating double-stranded RNA in the presence of CrfA but not CrfA-M2. We attribute this alternative pattern to the duplex formation between SisA and CrfA.
- **In further support of this duplex formation, we performed an EMSA with labelled SisA in the absence or presence of Hfq and CrfA wild-type or M2 mutation in the interaction site, respectively. In this experiment, we recover complexes of Hfq with wild-type SisA and CrfA, but not CrfA-M2. The EMSA was added as Fig. S6d.**
- **An *in vivo* study involving a compensatory exchange in CrfA/SisA (see Fig. 3c/d and previous comment) further supports a direct base-pairing between the two sRNAs.**
- **A combination of M1 versions of CrfA and SisA will restore sponge activity of CrfA on SisA (see 3c/d) but not target regulation, as the RNA stretch within SisA that interacts with CrfA is also required for base-pairing with the target.**

Minor comments:

1. Figure 1: The numbers to the right of staples in panels a and b are neither explained in the figure nor in the legend. I reckon this is the number of chimeras, but it should be clearly stated. In a, the writing “2.5 E05” may be unclear for some readers. Please use a more mathematically correct writing instead. In d, the font size for chromosomal positions needs to be increased.

The numbers in panel a depict the number of reads, in panel b the number of unique chimeras. We have replaced the scientific notation to decimal notation.

2. Sentence starting on line 131 “In addition to...” should include a reference to a figure or table.

This section has been restructured, and we added a reference to Table S1 listing all significant and high confidence chimeras.

3. Fig. 2b: Please add numbers in the structure representation for SisA. This is required for the reader to be able to evaluate the results of Fig. S6. Alternatively, add the structure with numbering next to the gel in Fig. S6.

We have added the secondary structure of SisA with numbered positions as Fig. S6b.

4. Line 164 and Fig. 2c: The text states that the M2 mutation replaces residues G49-G54 with “a non-complementary tetraloop sequence”. By contrast, the annotation in Fig. 2c gives the impression that those nucleotides simply were deleted (delta sign). Please clarify.

The figure has been updated.

5. Lines 179-180 and Fig. 3d: The text claims that deletion of Hfq abrogated the inhibitory effect of CrfA and SisA on each other. However, Fig. 3d shows practically no difference in CrfA levels upon SisA overexpression with respect to the presence or absence of Hfq. In both cases, SisA overexpression leads to a significant reduction of CrfA. Please change the text to properly describe the presented data.

We have rephrased this section.

6. Figure 5: I would highly recommend the authors to change the heat-map representation to a bar graph. This would make it easier to evaluate the differences between CrfA overexpression in “WT” and *sisAD* deletion strains.

We think that a heat map is the most suitable way to present this large dataset.

To facilitate comparison of individual expression levels, we now additionally supply all values presented in Figure 5 in Supplementary Table S2.

7. Line 307: the wording “the change in routine resulted in...” is unclear. Please rephrase to precisely describe what is meant.

We appreciate this feedback and have adjusted the phrasing of this sentence to clarify how the addition of xylose as the carbon source during recovery from starvation affected the *crfA* mutant strain.

We would like to thank the reviewers for the careful evaluation of our manuscript. We have addressed all comments and highlighted changes in the manuscript.

RESPONSE TO REVIEWER COMMENTS

Reviewer #1 (Remarks to the Author):

This manuscript presents a well-executed study on RNA-based regulation in *Caulobacter crescentus*, revealing how the carbon starvation-induced small RNA CrfA functions as a sponge to inactivate a family of sibling sRNAs (SisA-D), thereby reprogramming gene expression in response to changing nutrient availability. While previous work had identified CrfA as an important sRNA regulator, it was assumed to directly pair with and activate a set of targets involved in carbon transport. Using RIL-seq, in which pairs of sRNAs and pairing target mRNAs are ligated to identify chimeric RNAs, reflecting global RNA-RNA interactions, the authors show that CrfA, instead of making chimeras with the targets, was found to base-pair with a set of similar sRNAs, the most abundant of which is SisA. SisA (and the other Sis RNAs) pair with and negatively regulate the apparent CrfA targets; CrfA activates by relieving SisA repression. The data convincingly demonstrate that this regulatory mechanism is essential for *Caulobacter*'s adaptation to fluctuating carbon sources, with loss of CrfA leading to severe growth defects under stress. The manuscript is very well written, with a clear and engaging narrative. The background is thorough yet accessible, and the conclusions are strongly supported by the results. The issues raised below are to provide a bit more clarification and suggest a couple of additional tests.

Specific issues:

1. Fig. 1a: Is it useful to show this distribution for all RNAs, not just significant ones? This might be moved to supplemental material. While it is clear that chimeras are enriched significantly by the 3xFlag-Hfq vs. WT, (100x up), the singles are much less increased. Is this as high on WT as one would expect?

We have decided to show the distribution of all RNA species to outline our data analysis pipeline. As observed by the reviewer, chimeric reads (~70x) are twice more strongly enriched compared to single reads (~35x). We do not know whether that reflects any differences e.g. in stability of RNA species associated with the specific purification of Hfq-bound RNAs vs. the unspecific recovery on untagged Hfq protein in the wild-type sample but we have observed a similar discrepancy for a RIL-seq experiment in *Klebsiella pneumoniae* performed in our lab (Ruhland et al. 2024; PMID 38377209) but also in independent RIL-seq studies performed by other groups in *E. coli* (Melamed et al. 2016; PMID 27588604) or *S. Typhimurium* (Matera et al. 2022; PMID 35063132), suggesting that this phenomenon is a typical feature of the methodology and not specific to our study.

2. Lines 113-121: Of the 1535 statistically significant chimeras, only about half (607) were retained by the Chimeric Fragments pipeline. What gets left out by this? It might be useful to have the full list for reference (all statistically significant) for readers to have those to consider, understand what is removed by this program, which seems to require pairing. I don't think this program has been used in other RIL-Seq datasets. Are the orientations similar as the full set in terms of RNA1 vs. RNA2? Anything interesting in terms of what was left out? Is having 146/607 with at least one sRNA what one would expect?

The ChimericFragments pipeline (Siemers et al.; PMID 38633425) assigns an additional significance value based on the complementarity of the two RNAs in a chimeric read around the ligation point considered to be in proximity to the likely interaction site. In addition to benchmarking in the original publication, ChimericFragments was also applied to RIL-seq datasets for *Vibrio cholerae* ProQ (Ghandour et al. 2025; PMID 39727155), and to Hfq RIL-seq in *K. pneumoniae* (Ruhland et al. 2024; PMID 38377209). By applying this workflow to our dataset, we filtered for high confidence chimeras with good base-pairing predictions. This excluded mainly mRNA-mRNA and mRNA-IGR chimeras, as well as some sRNA-mRNA chimeras, many of which involved sRNAs antisense to transposases. The 928 chimeras that were filtered out also included 43 additional SisA chimeras and 7 CrfA-mRNA chimeras, 6 of which had the sRNA at position 1 as in its chimeras with SisA-D. The full list of statistically significant chimeras was added as an additional sheet to Table S1.

For high confidence chimeras, the distribution of RNA classes for position 1 and position 2 barely changes compared to the distribution of all significant chimeras (s. Fig. 1), e.g. 27% of sRNAs involved in high confidence chimeras are at position 1, which is a small decrease compared to 31% of sRNAs in all significant chimeras at position 1.

Figure 1: Relative distribution of RNA classes in significant chimeras (left) and high-confidence chimeras (right).

Around 25% of high confidence chimeras involve at least one sRNA. This fraction is smaller compared to what has been recovered in *E. coli* (40%, Melamed et al. 2016; PMID 27588604) or *K. pneumoniae* (47%, Ruhland et al. 2024; PMID 38377209), but slightly higher than in *Clostridioides* (20%, Fuchs et al. 2023 EMBO J; PMID 37140366). It is therefore likely that these differences are a species-specific characteristic.

3. Table S1: Please label CrfA as well as the other Sis genes in this table. For other entries that are listed as ncRNAs in position 1, it was sometimes difficult to understand what they are. Where would that be listed? For instance, 3_CCNA_02328; is this 3' UTR of gcrA? For others (for instance R0210), it is listed in NCBI as antisense and is shown with a pair that is antisense to that. Should that be indicated someplace as a category? In any case, an appropriate place to find all of these entries should be indicated in the table legend.

Thank you for this feedback on the labels and annotations in Table S1. The nomenclature was based on the current NCBI annotation of the *C. crescentus* NA1000 genome. We have renamed the

original CCNA_R0xxx annotation as SisA-D and CrfA. We clarified the 3_CCNA_xxxx nomenclature as 3'-derived sRNAs in the table legend and added a column indicating 3'-derived and antisense sRNAs.

4. Given the large number of SisA, CrfA chimeras, what would sRNA 1 or 2 look like aside from these? All of CrfA are in position 1; are almost all position ones that are ncRNAs pairing with another ncRNA (other decoys?)?

Even though the CrfA-SisA interaction is the most abundant regarding the number of reads, it is represented only by one unique chimera. Since our distribution analyses for Fig. 1b/c and S1c are based on the count of unique chimera and not the number of reads of those chimeras, excluding the CrfA-SisA interaction does not change the relative distribution of sRNAs at position 1 or position 2.

When examining our high confidence chimeras for possible other decoys, we find 41 chimeras with 27 different sRNAs in position 1. Most of the chimeras recovered with sRNAs in position 1 have an mRNA (22 of 41) or IGR (8 of 41) transcript mapped at position 2. Of the 27 sRNAs, only five exclusively have chimeric interactions with other sRNAs, adding up to 11 sRNA-sRNA chimeras for sRNAs in position 1 (27%). This includes CrfA (chimeras with SisA-D), the pair R0078-R0210 (in both orientations) with R0210 encoded as an antisense RNA to R0078, R0018 in a chimeric interaction with the partially overlapping antisense encoded R0019, and R0188 in a chimera with SisA. Whether these sRNA-sRNA interactions suggest decoy activities beyond typical antisense sRNAs has to be experimentally addressed.

5. Fig. S3e: The alignment of the upstream region of these genes appears, as noted to have nothing in common. Is there no core consensus sequence for *Caulobacter* anywhere upstream of these sRNA starts (with processing afterwards)? While they could have different activators or repressors, it seems surprising not to have any promoter regions.

As pointed out by the reviewer, a common binding site for the polymerase should be detectable at a distinct site upstream the transcription start site.

The motif for RpoD-binding in *Caulobacter* is relatively weak (see Figure; PMID 25569173); however, we do find the TTG motif at the expected distance (centred at -35) from the TSS for all sibling sRNAs. We have updated our alignment to highlight this information (see updated Figure S3).

6. Fig. S4 expression/levels: The results reinforce the idea that these different sRNAs have somewhat different patterns, presumably in part due to synthesis levels and in part degradation. In particular, does the pattern look very different in the absence of CrfA ?

All four sibling sRNAs have very distinct expression patterns, and as stated in the manuscript we think that different transcription factors are involved in their synthesis. CrfA likely plays a role in the decay of the Sis family RNAs and its deletion does mildly increase the expression maxima under standard conditions compared to the wild-type (see new Fig. S4b).

7. Fig. 2b-d: Please check legends for c, d (appear to be mixed up)

Thank you for pointing out this mistake, we have adjusted the legend for Fig. 2c-d.

8. Fig. 3: A bit more text to discuss these results would be useful. Does loss of Hfq have less effect on sRNA levels in *Caulobacter* than seen in some other organisms? Fig. 3d measures CrfA levels, which do still seem to decrease without Hfq when SisA is expressed (last bar significantly lower than control). SisA is more affected by CrfA, but in this case, only in hfq+. Is it surprising these are not reciprocal?

Compared to other Gram-negative species like *E. coli*, we observe on average less dramatic destabilization of sRNAs in the absence of Hfq in *Caulobacter* for the examples we have addressed (e.g. Vogt et al. 2024; PMID 38511926).

We have carefully rephrased this section (see also comment 5 of reviewer 4) as there is a reduced but detectable regulation in the absence of Hfq. We have newly added an EMSA which also indicates complex formation between CrfA and SisA in the absence of Hfq, although less efficient when compared to the reaction containing the protein (new Figure S6d).

We currently have not addressed the fate of CrfA and SisA when interacting with each other, and further studies will be needed to understand the competition between CrfA and the mRNA targets of SisA. We now added a comment regarding these points to the discussion.

9. Fig. 4: This is key experiment in demonstrating the epistasis, showing that SisA is needed for CrfA-dependent regulation, but not vice versa. Considering spelling this out a bit more.

We have added additional information on how CrfA was initially misinterpreted to emphasize the respective role of SisA and the sponge.

10. Fig. 4b, c: Is it necessary to use Hfq from *Caulobacter*? If so, that might be worth mentioning. *E. coli* Hfq is still present, correct? Hfq from *Caulobacter* overexpressed (please clarify in legend)? As above, this shows that SisA is sufficient for regulation. It would be good to also test CrfA. Presumably it does nothing? This would further confirm the epistasis in 4a.

We have previously shown that *Caulobacter* Hfq binds RNA with lower affinity and different specificity compared to *E. coli* Hfq (PMID 31076551). For our target validation experiments in *E. coli* we therefore used a strain that expresses *Caulobacter* Hfq as the sole copy under control of the endogenous *E. coli* *hfq* promoter (PMID 31076551). In addition to the methods section we have now added the description of the strain in the figure legend of Fig. 4c.

We agree with the reviewer that it is a good idea to test the effect of CrfA on the fusion.

Interestingly, the CrfA transcript fails to terminate efficiently in *E. coli*. We detect a run-through product terminating only at the internal terminator included on the plasmid (not shown) which is significantly longer than the sRNA to be tested. To overcome this issue, we have constructed a

version in which we attached a self-cleaving ribozyme to the 3' end of *crfA* on the *E. coli* expression plasmid. With this construct, we minimize run-through and produce full-length CrfA. We show that, as expected, CrfA in *E. coli* does not affect target reporter expression which confirms that SisA alone directly regulates expression of *CCNA_03574* (data shown in new Figure 4d; sRNA expression shown in new Figure S8).

11. Fig. 5: Overexpression of CrfA seems to have some effect (although much weaker), on the same genes in the absence of any sis genes. Is there an explanation for this? Another Sis-like mRNA, or possibly a second role of CrfA (by pairing or maybe encoding a short, active peptide)? Any reason to believe CrfA might bind/interfere with other roles of Hfq? Do any of the mRNAs that still show some CrfA effect have non-Sis chimeras? While this may all be beyond the scope of this work, it seems possible that such a biologically important RNA might be doing more things than the Sis decoy activity.

We agree with the reviewer that CrfA and the Sis sRNAs have ample potential for future studies. We currently have no evidence to support the hypothesis that CrfA may serve additional roles besides the regulation of the Sis sRNAs. A preliminary bioinformatics analysis of *crfA* conservation suggests that the sponge RNA is only present in species with at least one Sis RNA (unpublished). However, it is indeed intriguing that some transcripts were mildly deregulated in response to CrfA pulse-overexpression also in the absence of SisA-D. One possible explanation is an indirect effect on Hfq availability as the strong pulse induction of CrfA in this experiment could at least partially titrate the RNA chaperone, resulting in a short-term deregulation of the mRNAs. We would expect that the cell is able to balance this effect with time as we do not detect it in continuous overexpression. Indeed, the top four upregulated mRNA transcripts in response to CrfA pulse overexpression (*CCNA_03181*, *CCNA_03263*, *CCNA_03574*, *meaA* and *malA*), were significantly downregulated by SisA while CrfA had no effect on GFP levels (*03181* (not shown), *03263* (s. Fig. S9A), *03574* (s. Fig. 4)) or only caused mild repression (*meaA*, 1.6-fold (not shown)). Regulation of *malA* was not addressed.

In regard of the other questions in this comment:

- A ribosome-profiling experiment (PMID 25078267) did not reveal translation of the *crfA* transcript, we therefore currently exclude the possibility that CrfA is a dual function RNA. We cannot exclude the interaction of CrfA with another, currently unknown RBP but Hfq.
- Further high-confidence chimeras were found only between *meaA* and *CCNA_01805*, *malA* and *CCNA_03360*, and *malA* and *CCNA_02458*, respectively. We currently do not know whether these mRNA-mRNA pairs have any regulatory role.

12. Fig. 7: Is it known what would grow up (what suppressors, for instance) in *del crfA* in xylose? Are the transporters for xylose known?

We currently have not selected for mutants that enhance growth in the presence of xylose in a *crfA* deletion background. One could speculate that the inactivation of the xylose transport would rescue the phenotype but the uptake of this sugar is most likely not linked to a single gene/system. When Stephens et al. (2006) identified genes required for xylose utilization in *C. crescentus*, the transposon mutagenesis did not reveal a xylose transporter, indicating a potential redundancy in sugar uptake (PMID 17172333). In a follow-up, the authors showed that expression of XylE, a

predicted protein similar to the *E. coli* xylose:H⁺ symporter, was under control of the XylR repressor that regulates expression of the enzymes required for xylose metabolism *xylXABCD* (2007, PMID 17933895). Furthermore, *C. crescentus* encodes more than 60 TonB-dependent receptors (TBDRs) for which the individual substrates are currently mostly unknown. The uptake of xylose and xylo-oligosaccharides through TBDRs has been demonstrated for other bacteria, and it is therefore well possible that one or more transporters of this type also contribute to xylose uptake in *C. crescentus*. For example, the two TBDRs CCNA_01051 and CCNA_02923 are induced in the presence of xylose, however their function is yet to be determined (PMID 14973021).

13. Why was OD660 used some places, OD600 others?

The optical density of *Caulobacter* is typically measured at 660 nm, while *E. coli* optical density is measured at 600 nm. Thus, we use OD₆₀₀ in the legend of Figure S7b describing an experiment performed in *E. coli*.

14. Some suggestions for the discussion:

a. A bit more discussion of how CrfA likely interacts with Hfq and serves as a decoy or sponge would be useful. CrfA doesn't have a polyU tail, so may be more like a target and was entirely at position 1. Possibly this could be explicitly mentioned.

We have added a remark in the discussion to comment on the position of CrfA on Hfq as well as the potential competition with mRNA targets of the sibling sRNAs.

b. I would be inclined to more clearly discuss how CrfA was originally mis-interpreted (I think mostly that the pairing wasn't so clear, and the appropriate compensating mutations never tested?).

As assumed by the reviewer, the mechanism of positive regulation of CCNA_3574 mRNA (or any of the other transcripts identified to be upregulated) by CrfA had not been addressed in the original publication on this sRNA, and the potential base-pairing was not confirmed by a compensatory exchange. The originally used CrfA mutant was predicted to interrupt mRNA recognition but instead is positioned within the sibling sRNA binding site, and therefore failed to confer upregulation. We have added information into the discussion.

c. The authors do not currently refer to Figure 8 (the model figure) in the main text.

Thank you – we have fixed that issue and now refer to the figure.

d. Adding subheadings in the Discussion section could help structure the interpretation of findings and improve clarity for readers.

We did not include subheadings in the discussion as per Nature Communications guidelines.

Reviewer #2 (Remarks to the Author):

Reviewer #3 (Remarks to the Author):

In this study, Vogt and Velasco-Gomariz et al. examined the RNA-RNA interactome associated with the *Caulobacter crescentus* Hfq RNA chaperone protein using the RIL-seq approach. The authors detected many interactions between small RNAs (sRNAs) and other RNAs, of which a large percentage include the SisA sRNA. Most importantly, the Vogt and Velasco-Gomariz et al. found that the CrIA sRNA previously found to affect metabolism in response to carbon-starvation does not directly base pair with the reported targets but instead serves as a sponge RNA to downregulated constitutively-expressed SisA and the other aR8 RNA family RNAs SisB, SisC and SisD. Consistent with the CrIA role in regulating carbon metabolism through SisA, strains lacking the sponge RNA had growth defects in environments with fluctuating carbon levels.

The CrIA, SisA, SisB, SisC and SisD are interesting sRNAs, and the data are convincing. I have relatively minor comments:

1. The first part of the manuscript jumps around unnecessarily. Some interrelated suggestions:
--I suggest that the "story line" of the paper be about how CrfA was mischaracterized. The RIL-seq data showed that CrfA is a sponge!
--Considering the first point, there should be more information about CrfA as well as about sponge RNAs in the introduction. The finding that CrfA is a sponge RNA should be stated explicitly in the introduction. Similarly, I think the first few sentences of the results describing published findings make more sense in the introduction. Instead, the long paragraph about the transcriptional regulators, which do not feature in the results should be condensed.

As suggested by the reviewer we have added a section on RNA sponges to the introduction and cut the paragraph in the beginning of the results. We have also shortened the paragraph on *Caulobacter* sRNAs.

2. The Sis family of RNAs is intriguing. Given this the authors should:
--Page 5, line 141: Define and describe the aR8 RNA family at the first mention.

We are not entirely sure how to interpret this comment. We added a remark to the introduction to point to the affiliation of SisA-D with the aR8 family. The description and definition of SisA-D is at the indicated position.

--Include more of the findings about SisB, C and D (such as northern analysis) in the main figures.

We have added a Northern blot analysis on the differential expression of the Sis family to main Figure 2.

3. The authors should be explicit about experimental details such as whether the CCNA_03574::*gfp* is on a plasmid or the chromosome (Page 7, line 195).

We apologize if our description was misleading. Our reporter system in *Caulobacter* consists of a plasmid-encoded sRNA and a target mRNA::*gfp* fusion. The latter construct is always integrated

into the same, neutral locus on the chromosome (*rsaA* gene; not in the target gene). We have added a remark to the figure legend and the method section.

4. Some wording could be “toned down” without detracting from the paper.

--Page 2, line 27: sRNA-sRNA pairs or sponge RNAs are no longer so “unconventional”.

Done.

--Page 13, line 403: Given the findings described, the Qrr family no longer “has a peculiarity”.

Done.

5. Additional editorial suggestions:

--Page 3, line 50: “destined”? Perhaps replace with “utilized”.

Done.

--Page 4, line 103: Replace “co-immunoprecipitated from Hfq” with “co-immunoprecipitated with Hfq”.

Done.

--Page 5, lines 119-121: The numbers here are somewhat confusing. Are the authors referring to one third of 607 or of 146?

Done.

--Page 5, lines 131-133: The sentence beginning “In addition to these mRNA targets,...” is a little awkward.

Done.

--Page 6, line 145: Does CCNA_00780 encode a glutamate amidohydrolase?

Yes. We now emphasize that.

--Page 10, lines 319-321: Suggest “..the functional context in which distinct sRNAs are embedded⁴⁴. Here, we report RIL-seq mapping of the Hfq-dependent RNA-RNA network in *C. crescentus*.”

Done.

--Page 11, line 332: Replace “alternative” with “multiple”.

Done.

Reviewer #4 (Remarks to the Author):

In this manuscript, Vogt and co-workers employ the RIL-seq approach to globally identify RNA-RNA interactions associated with the Hfq protein in *Caulobacter crescentus*. From the hundreds of identified interactions, the authors focus on the association of the partially characterized sRNA CrfA and the four sequence-related sibling sRNAs SisA-D. The authors reveal that CrfA acts as a sponge of the Sis sRNAs, thereby relieving Sis-dependent regulation of a large number of mRNA targets connected to nutrient metabolism. The CrfA-Sis interaction is partially supported by mutational analysis in vivo and structure probing in vitro. SisA-dependent regulation of mRNA targets is convincingly determined through compensatory mutation experiments. Importantly, the authors show that loss of CrfA leads to a severe growth phenotype during recovery from carbon starvation.

This is a very well-written manuscript presenting data of high quality. It is an important contribution to the field, especially since it presents an example of sRNA-based regulation with strong impact on bacterial proliferation. Still, prior to publication, there are some issues that need to be addressed, as specified below.

Specific comments:

1. The RIL-seq data is based on four biological replicates. Please inform the reader about the reproducibility between the replicates, including overlap of chimeras and statistical correlation.

We have added information about the overlap between the individual replicates (Fig. S1d). The most abundant chimeras (like SisA-CrfA) are recovered by all four experiments. The first two datasets show less overlap, and one reason might be the lower sequencing depth that was chosen when the method was to be established in the lab. We have nevertheless decided to include results of all four experiments to maximize the number of interactions as a resource.

2. The authors convincingly show that CrfA-dependent regulation of mRNA translation, which previously has been suggested to be direct, in fact is indirect and through the Sis sRNAs (e.g. Fig. 4a). In Figure 3, the authors show that overexpression of CrfA leads to reduced SisA levels, and that overexpression of SisA leads to reduced CrfA levels. A point mutation in the predicted interaction site in either CrfA or SisA leads to reduced effect on the other sRNA. However, although the M1 mutants of either sRNA were designed to restore complementarity when combined, the authors did not show whether restoring complementarity also restores regulation. Thus, whether the two sRNAs indeed interact, and whether this interaction is required for regulation, still remains somewhat an open question.

As suggested by the reviewer we have now added an experiment to show that compensatory mutations in SisA and CrfA confirm regulation via a direct interaction of the two sRNAs (Fig. 3c/d).

3. A direct interaction between SisA and CrfA is further investigated in vitro using structure probing. The experiment shown in Fig. 2e show clear protection from lead cleavage of CrfA in the presence of Sis sRNAs. However, the reciprocal experiment where SisA cleavage is monitored in the presence of CrfA is less convincing (Fig. S6). The text referring to Fig. S6 reads “we detected base-pairing of CrfA to SisA....covering the conserved 10-mer sequence core”, but any information about how the result

in Fig. S6 supports this claim is lacking. In fact, there is no change in lead reactivity in the predicted CrfA binding site of SisA when comparing the presence or absence of SisA. Since the predicted binding site is in a region that according to Fig. 2b is single-stranded (nt 37-46 of SisA), CrfA binding in this region should result in clear protection from cleavage. Instead, there is a clear protection from lead cleavage around position 55 when Hfq is added, but this is not mentioned in the text. This not only casts doubt over the CrfA-SisA interaction, but also whether the structure representation shown in Fig. 2b is correct. Together, the *in vivo* and *in vitro* analysis of CrfA-SisA interaction needs to be more convincingly demonstrated. For instance, did the authors perform EMSAs with SisA-CrfA using the M1 mutants? Does co-expression of M1 versions of CrfA and SisA result in regulation of target mRNA translation, while WT/mutant combinations do not?

As this comment consists of multiple questions/remarks we have split our response.

- **In vitro experiments presented in 6a:**
We have now indicated the potential Hfq binding site in SisA in Fig. S6a. While lead cleavage of SisA in the presence of CrfA is mildly altered at the expected binding site we do see a clear change in the RNase III cleavage pattern with increased reactivity around C42 within the 10-mer sequence core, indicating double-stranded RNA in the presence of CrfA but not CrfA-M2. We attribute this alternative pattern to the duplex formation between SisA and CrfA.
- **In further support of this duplex formation, we performed an EMSA with labelled SisA in the absence or presence of Hfq and CrfA wild-type or M2 mutation in the interaction site, respectively. In this experiment, we recover complexes of Hfq with wild-type SisA and CrfA, but not CrfA-M2. The EMSA was added as Fig. S6d.**
- **An *in vivo* study involving a compensatory exchange in CrfA/SisA (see Fig. 3c/d and previous comment) further supports a direct base-pairing between the two sRNAs.**
- **A combination of M1 versions of CrfA and SisA will restore sponge activity of CrfA on SisA (see 3c/d) but not target regulation, as the RNA stretch within SisA that interacts with CrfA is also required for base-pairing with the target.**

Minor comments:

1. Figure 1: The numbers to the right of staples in panels a and b are neither explained in the figure nor in the legend. I reckon this is the number of chimeras, but it should be clearly stated. In a, the writing “2.5 E05” may be unclear for some readers. Please use a more mathematically correct writing instead. In d, the font size for chromosomal positions needs to be increased.

The numbers in panel a depict the number of reads, in panel b the number of unique chimeras. We have replaced the scientific notation to decimal notation.

2. Sentence starting on line 131 “In addition to...” should include a reference to a figure or table.

This section has been restructured, and we added a reference to Table S1 listing all significant and high confidence chimeras.

3. Fig. 2b: Please add numbers in the structure representation for SisA. This is required for the reader to be able to evaluate the results of Fig. S6. Alternatively, add the structure with numbering next to the gel in Fig. S6.

We have added the secondary structure of SisA with numbered positions as Fig. S6b.

4. Line 164 and Fig. 2c: The text states that the M2 mutation replaces residues G49-G54 with “a non-complementary tetraloop sequence”. By contrast, the annotation in Fig. 2c gives the impression that those nucleotides simply were deleted (delta sign). Please clarify.

The figure has been updated.

5. Lines 179-180 and Fig. 3d: The text claims that deletion of Hfq abrogated the inhibitory effect of CrfA and SisA on each other. However, Fig. 3d shows practically no difference in CrfA levels upon SisA overexpression with respect to the presence or absence of Hfq. In both cases, SisA overexpression leads to a significant reduction of CrfA. Please change the text to properly describe the presented data.

We have rephrased this section.

6. Figure 5: I would highly recommend the authors to change the heat-map representation to a bar graph. This would make it easier to evaluate the differences between CrfA overexpression in “WT” and *sisAD* deletion strains.

We think that a heat map is the most suitable way to present this large dataset.

To facilitate comparison of individual expression levels, we now additionally supply all values presented in Figure 5 in Supplementary Table S2.

7. Line 307: the wording “the change in routine resulted in...” is unclear. Please rephrase to precisely describe what is meant.

We appreciate this feedback and have adjusted the phrasing of this sentence to clarify how the addition of xylose as the carbon source during recovery from starvation affected the *crfA* mutant strain.

RESPONSE TO REVIEWER COMMENTS AFTER REVISION

Reviewer #1 (Remarks to the Author):

In this revision of a study of the small RNA network in *Caulobacter crescentus*, the authors have done a very nice job of answering questions raised by us and other reviewers, and have clarified what was done at various stages. The manuscript will be important both to those studying *Caulobacter* and how it responds to changing carbon sources and those interested in sRNA-based regulation; it also serves as a cautionary note on interpreting control by sRNAs.

Some minor issues that can easily be corrected:

1. Line 111: The total chimeric number 493,886 does not match what is shown in Fig. 1a (484,969). Clarification is needed.

The total number of chimeric reads in the 3xFLAG-Hfq strain depicted in Fig. 1a is the correct number. We have corrected the number in line 111 to 484,969.

2. Legend of Fig. S7a: CfrA-M1 (Pconst::crfA-M1) was not tested and should be removed from the legend.

We have corrected the Figure legend.

3. Figs. S7c/d were not mentioned in the main text. Either remove the figures or add them somewhere in the text.

Figures S7c-d were removed.

Reviewer #2 (Remarks to the Author):

Reviewer #4 (Remarks to the Author):

The authors have convincingly addressed all comments.